# UI-Vision: A Desktop-centric GUI Benchmark for Visual Perception and Interaction

**Shravan Nayak** [* 1 2 3]  **Xiangru Jian** [* 4 3]  **Kevin Qinghong Lin** [5]  **Juan A. Rodriguez** [1 3 6]  **Montek Kalsi** [3]
**Nicolas Chapados** [3]  **M. Tamer Özsu** [4]  **Aishwarya Agrawal** [1 2 7]  **David Vazquez** [3]  **Christopher Pal** [1 8 3 7]
**Perouz Taslakian** [3]  **Spandana Gella** [3]  **Sai Rajeswar** [3 1 2]

🌐 https://uivision.github.io

## Abstract

Autonomous agents that navigate Graphical User Interfaces (GUIs) to automate tasks like document editing and file management can greatly enhance computer workflows. While existing research focuses on online settings, desktop environments, critical for many professional and everyday tasks, remain underexplored due to data collection challenges and licensing issues. We introduce **UI-Vision**, the first comprehensive, license-permissive benchmark for offline, fine-grained evaluation of computer use agents in real-world desktop environments. Unlike online benchmarks, **UI-Vision** provides: (**i**) dense, high-quality annotations of human demonstrations, including bounding boxes, UI labels, and action trajectories (clicks, drags, and keyboard inputs) across 83 software applications, and (**ii**) three fine-to-coarse grained tasks—**Element Grounding**, **Layout Grounding**, and **Action Prediction**—with well-defined metrics to rigorously evaluate agents' performance in desktop environments. Our evaluation reveals critical limitations in state-of-the-art models like UI-TARS-72B, including issues with understanding professional software, spatial reasoning, and complex actions like drag-and-drop. These findings highlight the challenges in developing fully autonomous computer-use agents. With **UI-Vision**, we aim to advance the development of more capable agents for real-world desktop tasks.

---

[*]Equal contribution  [1]Mila - Quebec AI Institute  [2]Université de Montréal  [3]ServiceNow Research  [4]University of Waterloo  [5]National University of Singapore  [6]École de Technologie Supérieure  [7]CIFAR AI Chair  [8]Polytechnique Montréal. Correspondence to: Shravan Nayak <shravan.nayak@mila.quebec>, Xiangru Jian <x2jian@uwaterloo.ca>.

*Proceedings of the 42^{nd} International Conference on Machine Learning*, Vancouver, Canada. PMLR 267, 2025. Copyright 2025 by the author(s).

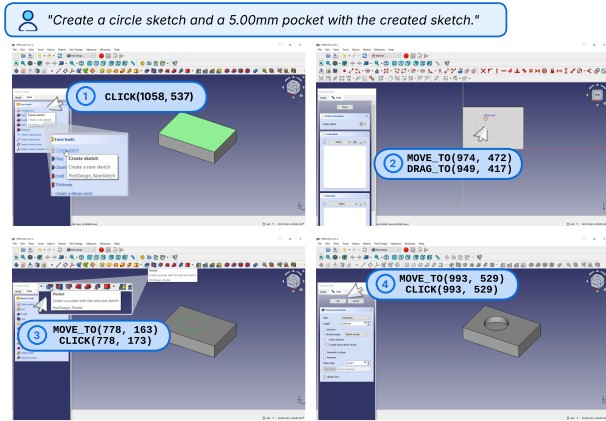

Figure 1: **UI-Vision's Action Prediction task.** Given a task instruction, a screenshot of the current UI state, and a history of previous interactions, the agent must predict the next action necessary to progress toward task completion. We show here an example of a successful episode from a task based on FreeCAD software[1].

## 1. Introduction

Graphical User Interfaces (GUIs) have become the dominant way users interact with digital worlds, replacing command-line interfaces with visually intuitive environments that enhance usability across desktops, web browsing, and mobile devices (Jansen, 1998). To accelerate digital workflows and assist users, there has been rapid progress in developing intelligent GUI agents—AI systems capable of automating GUI interactions, from simple tasks like auto-filling forms to complex operations such as configuring software settings. Recent advances in Large Language Models (LLMs) have significantly expanded the capabilities of these agents, allowing them to reason and follow natural language instructions (Wei et al., 2022; Yao et al., 2022b; Ouyang et al., 2022; OpenAI, 2021; Schick et al., 2023). However, LLMs that rely only on text struggle with GUI automation, as they lack the ability to interpret visual layouts, spatial relationships, and non-textual UI elements like icons (Gou

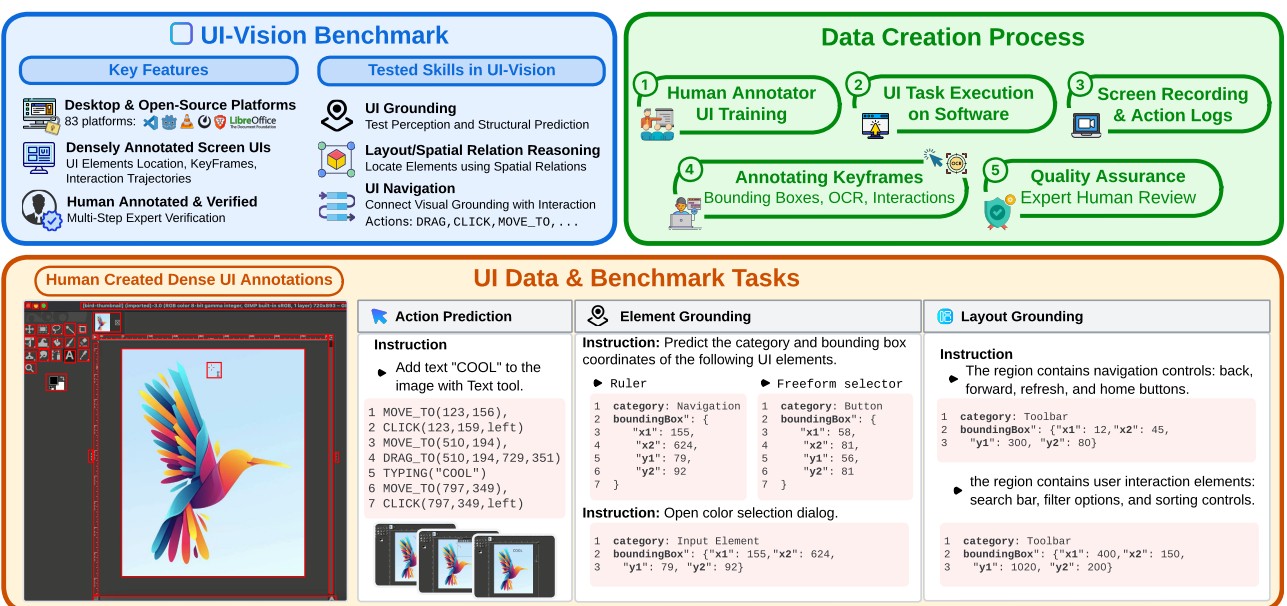

Figure 2: **UI-Vision Benchmark Overview.** Human annotators perform GUI tasks across various desktop software platforms, generating raw trajectories through screen recordings. The collected data undergoes expert verification and is then annotated with multiple layers of information, including keyframe detection, bounding box annotations with OCR, and action logs to capture user interactions.

et al., 2024b). Humans navigate software visually, recognizing elements by appearance and position. To replicate this, automation must incorporate vision, allowing models to process visual information beyond text. Multimodal LLMs have demonstrated effectiveness in this domain, successfully automating web browsing (Shi et al., 2017; Yao et al., 2022a; Gou et al., 2024a; Lù et al., 2024) and software control (Rodriguez et al., 2023; Li et al., 2023; Hong et al., 2024a; Qin et al., 2025), opening new possibilities for human-computer interaction.

Despite progress in LLM- and multimodal-driven automation, evaluating GUI agents for desktop remains challenging as existing benchmarks primarily focus on web and mobile environments. Web-based benchmarks like MiniWoB++ (Shi et al., 2017), Mind2Web (Deng et al., 2024), WebArena (Lù et al., 2024; Zhou et al., 2023) and WorkArena (Drouin et al., 2024) assess agent performance using Document Object Model (DOM) structures and HTML metadata. While effective for web interfaces, these approaches fail for desktop GUIs, which lack standardized text-based representations like HTML. Accessibility (A11y) trees provide a semantic alternative but are often inconsistent and incomplete, limiting their reliability for UI understanding (Gou et al., 2024a; Dardouri et al., 2024). Similarly, mobile-focused benchmarks (Li et al., 2020b; Toyama et al., 2021) emphasize touchscreen interactions, making them less applicable to desktop environments where interactions rely on precise mouse control and keyboard interactions. Furthermore, desktop applications lack standardized automation APIs,

unlike web automation, which benefits from tools like Selenium[2] and Playwright[3]. This lack of standardization hinders large-scale data collection and automation. Given these gaps, a dedicated desktop-centric benchmark is needed to evaluate models on real-world software variability, visual perception, and open-ended GUI interactions.

To bridge the gap in evaluating GUI agents for desktops, we introduce **UI-Vision**, a benchmark spanning 83 software applications across 6 domains designed to assess GUI agents' ability to perceive and interact with desktop environments in an offline setting. To build UI-Vision, participants design computer use tasks across these applications that reflect real-world software interactions. They record demonstrations for these tasks and annotate them by labeling all UI elements (Figure 13) and capturing action sequences such as clicking, dragging, and typing, creating a rich human-annotated dataset. For an agent to take meaningful actions within a GUI, it must first understand the interface it interacts with. This involves two key abilities: recognizing functional regions where UI elements are grouped together and identifying specific elements within these regions. Our evaluation framework builds on these foundational skills by assessing an agent's ability to perceive, interpret, and interact with a GUI through three core tasks. First, **Layout Grounding** evaluates how well an agent identifies functional groupings within a GUI, helping it understand the

---

[2]https://www.selenium.dev/
[3]https://playwright.dev/

broader structure of an interface (Figure 4 bottom right). Next, **Element Grounding** measures an agent's ability to precisely recognize and locate individual UI components (Figure 4). Finally, **Action Prediction** builds on these skills, testing whether agents can execute interactions like clicking, dragging, and typing based on their UI understanding (Figure 1). Unlike web and mobile interfaces, these tasks remain largely unexplored in desktop environments. UI-Vision fills this gap by providing a large-scale benchmark with extensive UI element coverage, spanning **83 platforms**, **450 recorded videos** with dense annotations, and **8267 query-label pairs**, creating a fine-grained and robust evaluation framework for multimodal GUI agents.

We evaluate state-of-the-art GUI agents on our UI-Vision benchmark, which reveals significant gaps across all three tasks. Element Grounding, a core ability for performing actions on GUI, remains particularly difficult where even the best-performing model, UI-TARS (Qin et al., 2025), achieves only 25.5% accuracy. This becomes more challenging when an interface is functionality-rich and visually complex, with an increased number of UI elements. Layout Grounding remains a challenging task, with Gemini-1.5-Pro (Team et al., 2024) achieving only 30.8 IoU, indicating that models lack high-level visual comprehension to recognize structured UI regions. Lastly, Action Prediction is consequently challenging, with UI-TARS achieving only 19.7% recall on click actions. The lack of accurate grounding naturally limits action prediction, as models struggle to associate UI elements with the correct interactions. Moreover, all models struggle with drag actions, exposing weaknesses in handling motion-based interactions, a capability not typically required in web-based GUI tasks.

In summary, our major contributions include:

- We introduce UI-Vision, the largest and most diverse desktop GUI benchmark, spanning 83 real-world environments. It enables a comprehensive evaluation of models in an offline setting across three core tasks: Element Grounding, Layout Grounding, and Action Prediction.

- Our dataset provides dense annotations with unmatched UI element coverage, allowing models to be tested on spatial relationships and UI layout areas often overlooked in prior benchmarks. Built from **open-source and permissive data**, it ensures accessibility and reproducibility.

- We identify major weaknesses in state-of-the-art models across core GUI tasks. Grounding of elements and layout remains highly challenging, as models struggle with fine-grained understanding and lack proper recognition of UI regions and their significance. These limitations ultimately hurt action execution, where GUI agents fail at precise click and drag operations.

## 2. Related Work

### 2.1. GUI Agents

Recent research highlights the expanding capabilities of large language models (LLMs) beyond conventional linguistic tasks including their application in GUI environments. A key milestone in task-driven GUI navigation is MiniWoB++ (Shi et al., 2017), a web-based environment. Interest in this problem has been renewed with the emergence of large language models, which have demonstrated remarkable capabilities in orchestrating complex workflows autonomously (Yao et al., 2023; Yang et al., 2024a; Gao et al., 2023), driving progress in GUI automation. Current methodologies broadly fall into two categories.

**Pure Language Agents** extract UI metadata from HTML structures, accessibility trees, or OCR (Lee et al., 2023) and spatial-semantic models (Yang et al., 2023; Yan et al., 2023; Zheng et al., 2024; Lu et al., 2024). LLMs then process this structured data to generate task-specific actions. While flexible, these approaches rely on closed-source models and struggle with generalization, as real-world applications often operate on raw visual inputs rather than structured metadata, which is not always available.

**Multimodal Agents** attempt to overcome these limitations by using multimodal datasets, pairing images with textual descriptions (Hong et al., 2023; Cheng et al., 2024; You et al., 2024; Li et al., 2024; Gou et al., 2024a; Lin et al., 2024b; Yang et al., 2024b; Wu et al., 2024; Qin et al., 2025). This approach enables pixel-level element grounding and context-aware interface navigation. However, progress remains constrained by the lack of large-scale visual data and standardized benchmarks for real-world UI interactions in desktop environments. Our work fills this gap by introducing a benchmark tailored to desktop applications.

### 2.2. GUI Benchmarks

Existing benchmarks evaluate different aspects of GUI agent capabilities but remain fragmented in their focus. Element grounding benchmarks (Cheng et al., 2024; Li et al., 2025) test an agent's ability to locate UI elements like icons and text. Action prediction datasets (Deng et al., 2024; Rawles et al., 2023; Zhang et al., 2024) assess task understanding and next-step inference based on screenshots and history. However, layout grounding is often overlooked—current coordinate-based methods (Li et al., 2025) fail to capture structural relationships in GUI tasks (Wu et al., 2023; Rodriguez et al., 2024). Most benchmarks specialize in either grounding (Cheng et al., 2024; Li et al., 2025), navigation (Rawles et al., 2023; Deng et al., 2024), or understanding (Hsiao et al., 2024), lacking a comprehensive evaluation framework. Furthermore, GUI benchmarks are disproportionately skewed towards **Web** (Hao et al., 2011; Wu et al.,

| Benchmarks | Environments | | | Tasks | | | Statistics | | | Data Collection | |
|---|---|---|---|---|---|---|---|---|---|---|---|
| | Platform | # SW/App | Settings | Ground. (multi.) | Action | Layout | # Sample | Avg. Ele | Avg. Steps | Human | Open License |
| MiniWoB++ (Shi et al., 2017) | Web | N/A | Online | ✗ | ✓ | ✗ | 125 | N/A | 2.3 | N/A | N/A |
| Mind2Web (Deng et al., 2024) | Web | N/A | Offline | ✗ | ✓ | ✗ | 2,350 | 1 | 7.3 | ✓ | ✓ |
| AITW (Rawles et al., 2023) | Mobile | 159 | Offline | ✗ | ✓ | ✗ | 30,378 | 1 | 6.5 | ✓ | N/A |
| OmniAct (Kapoor et al., 2024) | Desktop, Web | 38 | Offline | ✗ | ✓ | ✗ | 9,802 | 18.6 | 1 | ✓ | N/A |
| OS-World (Xie et al., 2024) | Desktop | 8 | Online | ✗ | ✓ | ✗ | 369 | N/A | N/A | ✓ | ✗(Windows) |
| VideoGUI (Lin et al., 2024a) | Desktop | 11 | Offline | ✗ | ✓ | ✗ | 86 | N/A | 22.7 | ✓ | N/A |
| ScreenSpot (Cheng et al., 2024) | Desktop, Web, Mobile | N/A | Offline | ✓(✗) | ✗ | ✗ | 1,272 | 1 | N/A | ✓ | N/A |
| ScreenSpot-Pro (Li et al., 2025) | Desktop | 23 | Offline | ✓(✗) | ✗ | ✗ | 1,581 | 1 | N/A | ✓ | ✓ |
| **UI-Vision (ours)** | Desktop | 83 | Offline | ✓(✓) | ✓ | ✓ | 8227 | 71 | 7.3 | ✓ | ✓ |

Table 1: **Comparison of existing GUI benchmarks with our UI-Vision.** We evaluate their inclusion of desktop platforms, noting a gap in recent works. Additionally, we examine permissive licensing (Open License), highlighting that UI-Vision focuses on open-source platforms. Lastly, we assess the presence of relevant annotations, particularly those related to grounding, text, and OCR. **Multi.** refers to Multi-purpose query for grounding.

2023; Zhou et al., 2023; Koh et al., 2024) and **Mobile** (Deka et al., 2017; Li et al., 2020a;b; Toyama et al., 2021) platforms, primarily due to their accessibility. **Desktop environments, despite their significance in professional workflows, remain underexplored.** Existing desktop benchmarks (Xie et al., 2024; Bonatti et al., 2024) primarily focus on online interactions, while others are constrained by small-scale datasets (Kapoor et al., 2024; Lin et al., 2024a).

Table 1 compares existing benchmarks, highlighting the lack of comprehensive desktop datasets with diverse applications, annotations and real-world complexity. To address these gaps, we introduce **UI-Vision**, the **largest desktop-centric benchmark** to date, spanning **83 software environments** and featuring **rich human annotations**. With 8227 query-label pairs across three tasks and a broad software and domain coverage, UI-Vision establishes a new standard for evaluation in desktop environments.

## 3. UI-Vision

This section introduces UI-Vision, a large-scale benchmark for GUI navigation and visual grounding across 83 desktop applications. We describe the data collection and annotation in Section 3.1 and in Section 3.2 explain how these annotations are used to construct the UI-Vision benchmark tasks for model evaluation.

### 3.1. Data Collection

We first collect data from users interacting with desktop software to achieve a goal, capturing their actions and annotating critical elements. This involves recording user interactions, extracting action trajectories that document step-by-step decisions to achieve a goal, and labeling UI components with bounding boxes and functional descriptions. We partner with a data-sourcing company for this process, ensuring a diverse and well-trained annotator pool. Details about the annotators and their qualifications are provided in the Appendix A. Below, we describe the key steps

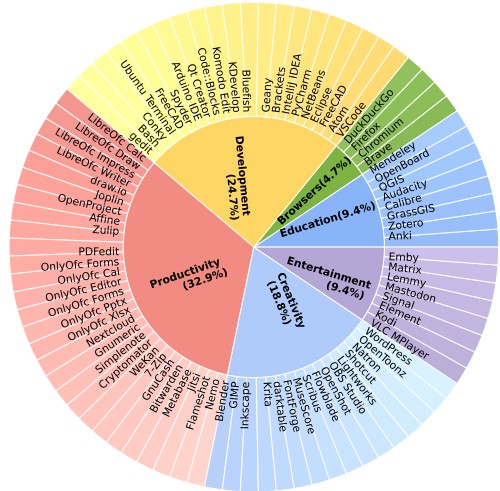

Figure 3: **Distribution of Software Platforms in UI-Vision.** The outer ring displays 83 different software platforms spanning six categories shown in the inner ring.

involved in data collection.

**Selecting Desktop Applications.** We curate data from 83 open-source desktop platforms across six domains: Productivity, Development, Creativity, Education, Browsers, and Social Media/Entertainment, ensuring diverse platforms for comprehensive benchmarking and permissive licensing (Figure 3 and Table 6).

**Designing Computer Use Tasks.** Computer use tasks are designed around real-world workflows, ranging from basic tasks (e.g., renaming a folder) to complex operations (e.g., applying subtitles to a video). We ensure that each task is well-defined, and comprehensive. Each platform includes 5–7 computer use tasks.

**Capturing and Annotating User Interactions.** Expert annotators perform computer use tasks while capturing (i) task videos, (ii) logged actions (10 predefined types; Table 7 in Appendix B) with metadata such as mouse click type, click frequency and exact timestamp of the action in the video,

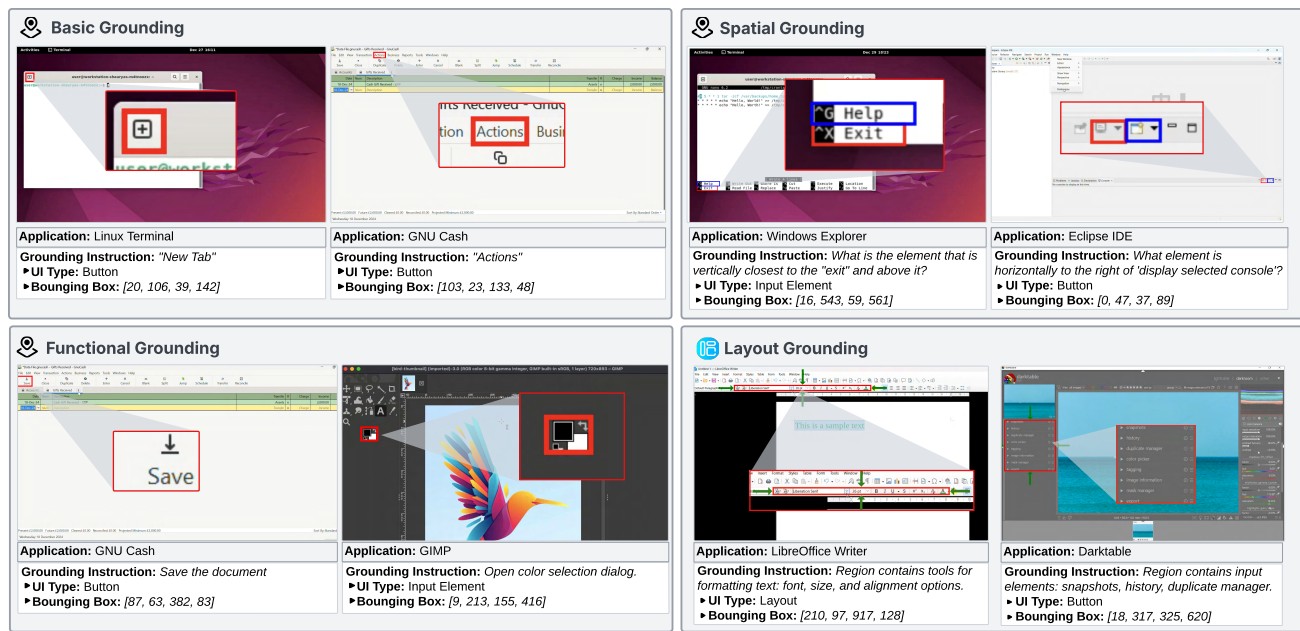

Figure 4: **Examples of Grounding Tasks in UI-Vision.** The dataset features three GUI grounding tasks: **Basic**, involving locating elements' bounding boxes; **Spatial**, focusing on determining positions relative to other elements; and **Functional**, requiring identification of click locations for specific actions. **Layout Grounding** locates larger regions given a description.

and (iii) keyframe screenshots, which are screenshots taken just before an action, such as a click, is performed. Post-task, annotators label UI elements with bounding boxes and descriptions for the keyframes using a proprietary tool. A multi-stage quality check by separate annotators and periodic verification by authors ensures completeness, accuracy, and consistency. The final dataset consists of 450 high-quality demonstrations across 83 applications. Figure 2 provides an overview of the data collection process.

**Dataset Statistics.** In Figure 3, we present the taxonomy and software distribution within UI-Vision, with the largest portions corresponding to 33.7% productivity, 25.3% development, and 19.3% creativity. Figure 12 in Appendix C presents key dataset statistics, including bounding box distribution per keyframe, task video durations, and action steps required for completion. Each keyframe contains between 5 to 200 labeled bounding boxes, with an average of 71, resulting in a dense annotation setup. This count is significantly higher than previous works (Table 1), highlighting the richness of our dataset. Most videos are under 50 seconds, averaging 37.6 seconds. Finally, annotators complete most tasks within 25 steps, with an average of 14 steps. This makes the dataset significantly complex, as agents must plan and predict actions over long-horizon tasks, which pose a major challenge.

### 3.2. UI-Vision Benchmark

Building on the rich annotations from the previous section, we focus on three key tasks essential for an agent's capa-

bilities: Element Grounding, to identify and localize UI elements from textual queries; Layout Grounding, to recognize structural relationships and group UI elements into functional clusters; and Action Prediction, to determine the correct interaction needed to achieve a goal. The first two address perception and structural prediction, while the third connects perception to interaction. See Figure 4 for examples of Element and Layout Grounding and Figure 1 for an example on the Action Prediction task.

TASK 1: ELEMENT GROUNDING

This task evaluates a model's ability to predict the bounding box of a UI component in a screenshot given a short query (e.g., "Add New Tab"). We leverage the dense bounding box annotations collected during the initial data collection stage to introduce three grounding subtasks—basic, functional, and spatial—to assess different aspects of GUI understanding beyond simple textual queries. To create these subtasks, we implement a multi-stage process that involves sampling challenging UI elements followed by thorough human evaluation of selected UI elements (see details in Appendix B.1). The basic setting evaluates a model's ability to predict the location of a UI element (e.g., button, sidebar, input field) given a minimal textual description, such as "Actions" (Figure 4). The functional setting requires identifying UI elements based on their function rather than direct labels; for instance, instead of "Save Button," the query specifies "Save the current document or data." Functional descriptions are generated using GPT-4o (OpenAI, 2023) and then validated by human reviewers. The spatial setting requires locating

| Action | Description |
|---|---|
| `Move(x, y)` | Move the mouse to the specified coordinates. |
| `Click(x, y, button)` | Click the specified button at the given coordinates. |
| `Typing('Hello')` | Types a specified string. |
| `Hotkey('ctrl', 'c')` | Performs individual or combination hotkeys. |
| `Drag([x1,y1], [x2,y2])` | Drags the mouse from start $[x1, y1]$ to end $[x2, y2]$. |

Table 2: **Action Prediction task**, action space.

UI elements based on their spatial relationships with neighboring components. Queries are generated by detecting the nearest bounding boxes in four directions (left, right, top, bottom), e.g., "What is directly above the 'Exit' button?" Both basic and functional grounding subtasks have 1,772 query-label pairs, while the spatial setting has 1,935.

TASK 2: LAYOUT GROUNDING 🔢

GUI layouts define how interface components, such as buttons, text fields, and containers, are arranged into cohesive regions (e.g., "Formatting Tools" or "Main Navigation Bar"). This novel task evaluates a model's ability to cluster UI elements into functional and semantic groups and predict bounding boxes that encapsulate them, a capability not explicitly explored in previous GUI-related benchmarks (Figure 4, bottom right). We again leverage the dense bounding box annotations collected during the initial data collection phase and provide them as input to LLAMA-3.3-70B (Meta, 2024), which generates non-overlapping functional or semantic clusters along with textual descriptions (e.g., "Formatting Tools"). These textual descriptions serve as queries for the model, which must predict bounding boxes that encapsulate entire functional or semantic areas. The dataset comprises 311 human-verified query-label pairs from 77 platforms, with each functional or semantic cluster containing 5–10 UI elements (see details in Appendix B.2).

TASK 3: ACTION PREDICTION 🚀

Action prediction builds on the previous perception tasks, requiring the agent to solve computer use tasks through a structured sequence of interactions. Unlike static recognition tasks, this agentic task evaluates a model's capability to interpret a given goal (e.g., "Apply a transparency of 45.9% to the layer") and determine the correct sequence of actions. We use the action trajectories collected during the data collection stage to create this task. Each step in the trajectory is treated as an independent prediction, with the model receiving the screenshot of the current UI state and previous actions as input. More concretely, given a task query $\mathcal{Q}$ and a screenshot $\mathcal{I}i$ at the $i$-th state, along with the previous action history $\mathcal{H} = \{a_j\}_{j \leq i-1}$, the model is required to output $\tilde{a}_i$. Each action $a_i$ includes both the action type and its parameters. The details of creating the task including choosing the right screenshots from videos for prediction and processing action trajectories are detailed in Appendix B.3. As a result, we obtain 3191 action-annotation pairs over 442 computer use tasks. In Table 2, we explicitly define

our action spaces and their corresponding parameters.

# 4. Experiments

## 4.1. Baselines

We tested several open-source VLMs with GUI capabilities, including Qwen2-VL-7B and Qwen2.5VL-7B (Wang et al., 2024), MiniCPM-V-2.6-8B (Yao et al., 2024), InternVL2-8B and InternVL2.5-8B (Chen et al., 2024); open source GUI agents, including CogAgent-9B (Hong et al., 2023), SeeClick-9.6B (Cheng et al., 2024), UGround-7B and 72B (Gou et al., 2024b), UI-TARS 7B and 72B (Qin et al., 2025), OSAtlas-7B (Wu et al., 2024), ShowUI-2B (Lin et al., 2024b), Aria-UI-25.3B(3.9B active parameters) (Yang et al., 2024b) and Aguvis-7B (Xu et al., 2024). We also considered closed source models such as GPT-4o (OpenAI, 2024), Claude-3.5-Sonnet and Claude-3.7-Sonnet (Anthropic, 2024), and Gemini-1.5-pro and Gemini-Flash-2.0 (Team et al., 2024). Each model used its recommended format for prompting.

## 4.2. Evaluation Metrics

**Element Grounding.** Following prior works (Li et al., 2025; Cheng et al., 2024; Wu et al., 2024), a prediction is considered *correct* if the predicted point $(x_i, y_i)$ falls within the ground-truth bounding box: Correct $= \mathbb{1}(x_{\min} \leq x_i \leq x_{\max} \wedge y_{\min} \leq y_i \leq y_{\max})$. We compute accuracy across all UI components. As prior analysis (Cheng et al., 2024) showed minimal gains from bounding-box-based evaluation, we adopt point-based evaluation throughout.

**Layout Grounding.** We assess predicted bounding boxes using **Intersection over Union (IoU)**, **precision**, and **recall**, measuring alignment with ground truth.

**Action Prediction.** Inspired by action evaluation metrics used by (Lin et al., 2024a), we develop metrics for each action type in our dataset:

• **Click & Move:** We evaluate coordinate predictions $[x, y]$ using two metrics: **Distance (Dist.)** – normalized Euclidean distance ($D$) between predicted and ground-truth locations: Dist $:= \frac{D}{L}$, where $L$ is the maximum possible distance to any corner. **Recall@$d$** – a click is correct if within $d$ pixels of the ground truth. We calculate $d$ as the average bounding box size across the dataset.

• **Drag:** Evaluated using: **Distance (Dist.)** measures average displacement error by computing the mean Euclidean distance between the predicted and actual start and end coordinates of the drag action: Dist $:= \frac{1}{2}\left(\frac{\Delta s}{L_s} + \frac{\Delta e}{L_e}\right)$. **Recall@$d$** checks if both start and end positions are within $d$ pixels.

• **Typing & Hotkey:** Evaluated using **Correctness**

| Model | Basic | | | | | | | Functional | | | | | | | Spatial | | | | | | | Final Avg |
|---|---|---|---|---|---|---|---|---|---|---|---|---|---|---|---|---|---|---|---|---|---|---|
| | Ed (215) | Br (56) | De (376) | Pr (605) | Cr (438) | En (82) | Overall (1772) | Ed (215) | Br (56) | De (376) | Pr (605) | Cr (438) | En (82) | Overall (1772) | Ed (212) | Br (31) | De (338) | Pr (740) | Cr (586) | En (28) | Overall (1935) | |
| *Closed-Source VLMs* | | | | | | | | | | | | | | | | | | | | | | |
| GPT-4o (OpenAI, 2024) | 2.23 | 0.00 | 1.86 | 1.16 | 1.14 | 4.88 | 1.58 | 1.40 | 0.00 | 3.19 | 0.83 | 0.91 | 3.66 | 1.52 | 0.94 | 0.00 | 1.48 | 1.22 | 0.51 | 3.57 | 1.03 | 1.38 |
| Gemini-1.5-pro (Team et al., 2023) | 0.47 | 0.00 | 1.60 | 0.83 | 0.46 | 0.00 | 0.79 | 0.00 | 1.79 | 0.27 | 0.17 | 0.46 | 0.00 | 0.28 | 0.94 | 0.00 | 0.89 | 0.54 | 0.34 | 0.00 | 0.57 | 0.55 |
| Gemini-Flash-2.0 (Team et al., 2023) | 0.00 | 0.00 | 0.27 | 0.66 | 0.68 | 0.00 | 0.45 | 0.47 | 1.79 | 0.00 | 0.66 | 0.23 | 0.00 | 0.40 | 0.00 | 0.00 | 0.00 | 0.14 | 0.00 | 0.00 | 0.05 | 0.30 |
| Claude-3.5-Sonnet (Anthropic, 2024) | 3.26 | 16.1 | 5.32 | 6.94 | 1.83 | 4.88 | 5.08 | 5.12 | 19.6 | 4.79 | 5.95 | 2.51 | 4.88 | 5.19 | 2.83 | 9.68 | 5.03 | 2.43 | 2.56 | 7.14 | 3.15 | 4.47 |
| Claude-3.7-Sonnet (Anthropic, 2024) | 6.51 | 12.5 | 7.98 | 11.24 | 9.13 | 11.0 | 9.48 | 5.12 | 7.14 | 8.24 | 9.92 | 6.16 | 4.88 | 7.73 | 6.60 | 9.68 | 7.69 | 7.43 | 7.85 | 10.7 | 7.60 | 8.27 |
| *Open-Source VLMs* | | | | | | | | | | | | | | | | | | | | | | |
| Qwen-2.5VL-7B (Wang et al., 2024) | 0.47 | 0.00 | 1.33 | 1.65 | 0.68 | 1.22 | 1.24 | 0.47 | 0.00 | 0.80 | 1.16 | 0.46 | 1.22 | 0.79 | 0.47 | 0.00 | 1.48 | 0.00 | 0.51 | 0.00 | 0.51 | 0.85 |
| InternVL2-8B (Chen et al., 2024) | 0.00 | 0.00 | 0.00 | 0.02 | 0.00 | 0.14 | 0.11 | 0.00 | 0.00 | 0.27 | 0.00 | 0.00 | 1.22 | 0.11 | 0.00 | 0.00 | 0.00 | 0.14 | 0.00 | 0.00 | 0.05 | 0.09 |
| InternVL2.5-8B (Chen et al., 2024) | 0.93 | 8.93 | 3.46 | 2.31 | 1.37 | 4.88 | 2.48 | 1.40 | 7.14 | 3.72 | 2.81 | 1.60 | 6.10 | 2.82 | 0.94 | 3.23 | 1.78 | 0.68 | 0.68 | 3.57 | 0.98 | 2.09 |
| Qwen-2VL-7B (Wang et al., 2024) | 2.79 | 7.14 | 3.72 | 3.97 | 0.68 | 12.2 | 3.44 | 2.79 | 12.5 | 3.19 | 3.97 | 0.68 | 6.10 | 3.22 | 0.47 | 3.23 | 2.37 | 1.08 | 0.51 | 3.57 | 1.45 | 2.70 |
| MiniCPM-V-8B (Yao et al., 2024) | 4.19 | 21.4 | 7.71 | 7.44 | 3.65 | 18.3 | 7.11 | 4.19 | 19.6 | 6.38 | 4.63 | 2.97 | 11.0 | 5.30 | 0.47 | 3.23 | 1.78 | 0.27 | 0.17 | 3.57 | 1.45 | 4.34 |
| *Open-Source GUI Agents* | | | | | | | | | | | | | | | | | | | | | | |
| ShowUI-2B (Lin et al., 2024b) | 5.12 | 16.1 | 9.84 | 9.09 | 3.42 | 19.5 | 8.07 | 5.12 | 12.5 | 9.31 | 8.60 | 4.11 | 15.9 | 7.67 | 0.94 | 9.68 | 2.96 | 2.70 | 0.68 | 3.57 | 2.07 | 5.94 |
| AriaUI-25.3B (Yang et al., 2024b) | 10.7 | 23.2 | 13.0 | 12.9 | 8.22 | 20.7 | 12.2 | 12.6 | 19.6 | 15.4 | 14.6 | 10.5 | 22.0 | 14.0 | 3.77 | 9.68 | 4.44 | 4.86 | 2.22 | 7.14 | 3.98 | 10.1 |
| UGround-v1-7B (Gou et al., 2024a) | 11.6 | 35.7 | 19.7 | 15.0 | 11.0 | 18.3 | 15.4 | 15.4 | 33.9 | 22.3 | 16.5 | 11.6 | 19.5 | 17.1 | 4.25 | 6.45 | 9.76 | 6.35 | 4.44 | 14.3 | 6.25 | 12.9 |
| OSAtlas-7B (Wu et al., 2024) | 10.7 | 23.2 | 13.3 | 12.6 | 8.22 | 22.0 | 12.2 | 11.6 | 16.1 | 11.4 | 12.7 | 7.53 | 13.4 | 11.2 | 3.77 | 6.45 | 5.62 | 3.65 | 2.22 | 7.14 | 3.67 | 9.02 |
| UGround-7B (Gou et al., 2024a) | 10.2 | 23.2 | 14.9 | 10.6 | 7.53 | 19.5 | 11.5 | 12.1 | 25.0 | 15.2 | 11.2 | 7.99 | 20.7 | 12.2 | 2.36 | 0.00 | 4.14 | 2.70 | 2.22 | 7.14 | 2.79 | 8.83 |
| Aguvis-7B (Xu et al., 2024) | 16.7 | 37.5 | 22.3 | 16.2 | 12.6 | 26.8 | 17.8 | 17.2 | 35.7 | 21.5 | 18.0 | 13.0 | 24.4 | 18.3 | 5.19 | 9.68 | 6.51 | 4.05 | 4.78 | 14.3 | 5.06 | 13.7 |
| UI-TARS-7B (Qin et al., 2025) | 15.4 | 41.1 | 21.8 | 21.2 | 13.2 | 39.0 | 20.1 | 20.5 | 41.1 | 25.5 | 26.5 | 16.0 | 45.1 | 24.3 | 6.60 | 12.9 | 11.0 | 9.19 | 5.80 | 17.9 | 8.37 | 17.6 |
| CogAgent-9B (Hong et al., 2024b) | 11.2 | 14.3 | 12.5 | 13.7 | 8.68 | 15.9 | 12.0 | 11.6 | 14.3 | 11.4 | 14.7 | 8.22 | 18.3 | 12.2 | 3.30 | 0.00 | 1.18 | 4.05 | 1.37 | 7.14 | 2.63 | 8.94 |
| SeeClick-9.6B (Cheng et al., 2024) | 7.44 | 23.2 | 13.0 | 8.43 | 5.48 | 17.1 | 9.42 | 4.65 | 7.14 | 5.32 | 3.97 | 4.34 | 7.32 | 4.68 | 0.47 | 6.45 | 3.25 | 1.22 | 2.73 | 3.57 | 2.07 | 5.39 |
| UGround-v1-72B (Gou et al., 2024a) | 27.0 | 42.9 | 31.7 | 26.6 | 22.8 | 40.2 | 27.9 | 25.1 | 33.9 | 30.6 | 26.6 | 21.0 | 40.2 | 26.7 | 15.1 | 25.8 | 19.8 | 13.4 | 12.8 | 25.0 | 14.9 | 23.2 |
| UI-TARS-72B (Qin et al., 2025) | 30.7 | 48.2 | 32.7 | 33.6 | 21.9 | 51.2 | 31.4 | 29.8 | 46.4 | 30.9 | 34.1 | 22.6 | 36.6 | 30.5 | 13.7 | 16.1 | 19.2 | 15.4 | 11.1 | 25.0 | 14.7 | 25.5 |

Table 3: Element Grounding results by category for Basic, Functional, and Spatial settings. For each setting, the first six columns (Ed, Br, De, Pr, Cr, En) present the fine-grained breakdown, followed by an overall score column. The number of samples in each category is noted under the column name. The final column shows the overall average. Abbreviated category labels: Ed (Education), Br (Browsers), De (Development), Pr (Productivity), Cr (Creativity), En (Entertainment). The best model within each size category is highlighted in **bold**, and the runner-up is underlined. Models are categorized by size: gray marks closed-source models, green indicates open-source VLM models, blue represents open-source GUI Agent models below 8B (active) parameters, orange denotes open-source GUI Agent models above (active) 8B parameters.

**(Corr.)**, which verifies exact string or keystroke matches.

We also compute the **Step Success Rate** (Deng et al., 2024) as an overall performance measure across the dataset. A step is considered successful only if the predicted action and its associated metadata are correct. This includes bounding boxes for click and drag actions, as well as keyboard inputs for typing and hotkey actions.

### 4.3. Results

**Performance on Element Grounding.** Table 3 presents model performance on the three Element Grounding subtasks. Both closed-source and open-source VLMs struggle with this task. The best closed-source model, Claude 3.7 Sonnet (Anthropic, 2024), achieves only 8.7% accuracy, while the best open-source model, MiniCPM-V-8B (Yao et al., 2024), performs even worse at 4.3%. This poor performance likely stems from these models being trained for broad visual understanding, including grounding objects in natural scenes, rather than the fine-grained grounding required for desktop environments, where UI elements are smaller and context-dependent. GUI agents trained on large-scale GUI data for UI understanding and grounding tasks perform best, with UI-TARS achieving 25.5% and UGround closely following at 23.2%. However, their overall perfor-

mance remains low, highlighting the challenges of grounding in desktop environments. Functional grounding yields similar scores to basic grounding for GUI agents, as they are explicitly trained on such functional tasks (Gou et al., 2024b; Qin et al., 2025). In contrast, both open-source and closed-source VLMs perform worse in this setting. The biggest challenge lies in spatial grounding, where even the best model achieves only 14.9%, underscoring the difficulty of understanding spatial relationships in GUIs.

A closer analysis reveals two major factors influencing model performance. First, as shown in Figure 5(a), models perform better when grounding larger UI elements, suggesting that smaller elements are harder to predict. Second, Figure 5(b) indicates that accuracy drops sharply when more elements are present in a screenshot. This pattern is evident across domains—creativity platforms such as Blender and GIMP exhibit the lowest performance, likely due to their densely packed GUI interface, with an average of 112 bounding boxes per screen and an UI element area averaging 418 square pixels. In contrast, entertainment platforms like VLC Media Player achieve the highest performance, benefiting from larger and more spaced-out UI elements, averaging 62.8 bounding boxes per screen and an average area of 875 square pixels per element. While it is commonly assumed

| | Layout Grounding | | |
|---|---|---|---|
| **Model** | **IoU↑** | **Precision↑** | **Recall↑** |
| *Closed-Source VLMs* | | | |
| GPT-4o (OpenAI, 2024) | 20.0 | 59.6 | 24.1 |
| Claude-3.5-Sonnet (Anthropic, 2024) | 22.4 | 64.3 | 26.8 |
| Claude-3.7-Sonnet (Anthropic, 2024) | 17.6 | 31.5 | 34.1 |
| Gemini-1.5-pro (Team et al., 2023) | **30.8** | **67.8** | 36.9 |
| Gemini-2.0-flash (Team et al., 2023) | 28.3 | 63.0 | 34.2 |
| *Open-Source VLMs* | | | |
| Qwen-2VL-7B (Wang et al., 2024) | 24.3 | 65.7 | 33.4 |
| MiniCPM-V-8B (Yao et al., 2024) | 16.3 | 25.7 | **43.6** |
| *Open-Source GUI Agents* | | | |
| CogAgent-9B (Hong et al., 2024b) | 6.22 | 7.99 | 42.9 |
| SeeClick-9.6B (Cheng et al., 2024) | 5.11 | 6.32 | 30.1 |
| OSAtlas-7B (Wu et al., 2024) | 28.2 | 66.4 | 41.6 |

Table 4: **UI-Vision Leaderboard on Layout Grounding**. We present the performance results of state-of-the-art UI understanding models that can predict bounding boxes. Results are averaged across IoU, precision, and recall.

that screenshot resolution significantly impacts grounding accuracy, our analysis (see Appendix E.1) reveals that it is not a key factor empirically.

**Error Analysis on Element Grounding.** Our analysis highlights three key challenges in model performance (see Appendix D.1). (i) *Fine-grained ambiguity:* Models often understand the query but struggle to distinguish the correct target when multiple similar-looking elements are present (Figure 14). (ii) *Lack of domain knowledge:* Agents fail to interpret certain symbols, such as the letter "F," due to missing context, suggesting the need for external knowledge sources like software documentation (Figure 15). (iii) *Small elements:* Models have difficulty recognizing small UI components (Figure 16), especially in high-resolution interfaces, indicating that strategies like iterative zooming (Wu & Xie, 2024) could improve performance. (iv) *Cross-platform generalization:* Models incorrectly apply UI layout assumptions from one platform to another, such as confusing button locations between Windows and iOS systems (Figure 17).

**Performance on Layout Grounding.** Table 4 summarizes the evaluation results on Layout Grounding. We only evaluate models capable of generating bounding boxes across closed-source VLMs, open-source VLMs, and GUI agents. Unlike Element Grounding, VLMs—both open and closed-source—outperform GUI agents in detecting UI layouts. Closed-source VLMs achieve state-of-the-art performance, while open-source VLMs perform comparably. This is likely due to their strong general visual understanding, which aids in semantic comprehension of the UI screen which is necessary for this task. Additionally, their training on grounding, even in natural image settings, may contribute to their success. In contrast, GUI agents perform poorly on this task. Both CogAgent-9B and SeeClick-9.6B show subpar results. While OSAtlas-7B is the exception, it is trained upon Qwen-

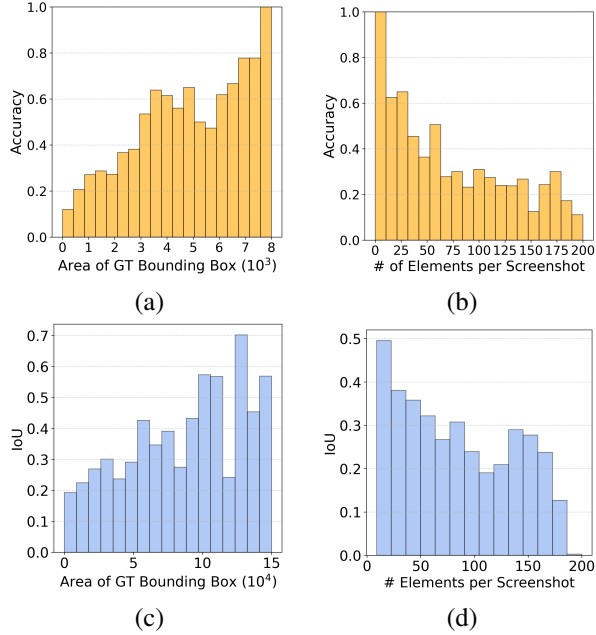

Figure 5: Analysis on Element Grounding (UI-TARS-72B under **Basic setting**) in terms of (a) Area of GT. Bbox and (b) # Element per screenshot. ; Layout Grounding (Gemini-1.5-pro) in terms of (c) Area of GT. Bbox and (d) # Element per screenshot. To enhance clarity, we do not display long-tail examples below 5%.

2VL and does not show significant improvement compared to the latter. This might be due to the fact that current GUI agent frameworks focus primarily on element grounding, such as improving click accuracy, rather than layout understanding and bounding box prediction.

Similar to grounding, IoU increases with the ground-truth bounding box area (Figure 5 (c)) and decreases as element density rises (Figure 5 (d)). Hence, complex and cluttered UI layouts make precise layout generation more challenging and remains the primary bottleneck in layout grounding.

**Error Analysis on Layout Grounding.** Models exhibit distinct failure patterns in layout grounding (see Appendix D.2). (i) *Inaccurate bounding box placement:* Closed-source models often fail to return the minimal bounding box for a target region, even when it is contained within their prediction (see Figure 18a, 19a). This suggests a weak understanding of layout partitioning. (ii) *Poor functional grouping:* Open-source GUI agents struggle to interpret UI structures at a higher level, failing even on queries that explicitly mention relevant elements (see Figure 18b). This indicates quite a weak understanding of functional groups with coarser granularity maybe due to a lack of such training data. (iii) *Superficial semantic matching:* When uncertain, open-source models tend to select smaller elements based on partial keyword matches rather than true semantic relevance, leading to incorrect predictions (Figure 19b). These limitations highlight

| Model | Click / Move | | Drag | | Typing | Hotkey | SSR |
|---|---|---|---|---|---|---|---|
| | Dist. ↓ | Recall@d ↑ | Dist. ↓ | Recall@d ↑ | Corr. ↑ | Corr. ↑ | ↑ |
| *Naive Baselines* | | | | | | | |
| Random | 81.6 | 0.0 | 94.2 | 0.0 | N/A | N/A | N/A |
| GPT-4o (w/o image) | 52.0 | 3.3 | 72.4 | 0.0 | 22.7 | 34.0 | 7.64 |
| *Closed-Source VLMs* | | | | | | | |
| GPT-4o (OpenAI, 2024) | 41.2 | 4.4 | 63.9 | 1.5 | **32.1** | **56.5** | 11.5 |
| Gemini-1.5-Pro (Team et al., 2024) | **38.7** | **13.0** | **61.1** | **1.6** | 24.7 | 45.3 | **16.0** |
| Claude-3.5-Sonnet (Anthropic, 2024) | 41.0 | 4.8 | 61.4 | 1.1 | 29.0 | 39.2 | 9.9 |
| *Open-Source GUI Agents* | | | | | | | |
| ShowUI-2B (Lin et al., 2024b) | **42.8** | 11.8 | N/A | N/A | 15.2 | **62.5** | 15.7 |
| UI-TARS-7B (Qin et al., 2025) | 47.0 | **19.7** | **64.8** | **3.1** | **33.8** | 40.5 | **21.4** |

Table 5: **UI-Vision Action Prediction Leaderboard**. We compare the performance of three model categories: Closed-Source VLMs, Open-Source VLMs, and Agentic solutions across four action types—Click/Move, Drag, Typing, and Hotkey. The evaluation metrics include Distance (Dist.) and Recall@d for Click and Drag actions, while Correctness (Corr.) is reported for Typing and Hotkey. Step Success Rate (SSR) represents the overall performance of models on the Action Prediction task.

the need for better spatial reasoning and structural awareness in GUI agents.

**Performance on Action Prediction.** Table 5 summarizes the results of the Action Prediction task. The random baseline, where actions and bounding box coordinates are selected at random for click and drag actions, performs very poorly, achieving 0% recall for both highlighting the challenging nature of the setup. In all our settings, we provide action histories to the model. To assess its impact, we set up an experiment where only task descriptions and action histories are given to GPT-4o without the image (GPT-4o (w/o image) in Table 4). The results suggest that action history only significantly improves performance compared to the random baseline. Notably, the model also performs well for typing and hotkey actions. This may be because prompts explicitly state information like emails or passwords for typing, while hotkeys sometimes follow predictable sequences, such as pressing the 'enter' key after typing.

Among closed-source models, Gemini-1.5-Pro outperforms GPT-4o and Claude-3.5-Sonnet on click and drag actions. Gemini's strong performance in the Layout Grounding task suggests that a high-level understanding of UI structures aids in predicting actions more accurately. In contrast, GPT-4o achieves significantly higher accuracy on keyboard-related actions. However, all models struggle with click and drag actions, indicating inadequate grounding—a trend also observed in the Element Grounding task. Moreover, poor drag action performance reveals a weakness in handling motion-based interactions, likely due to their limited presence in web-based tasks, which dominate training data.

Closed-source models are known for generating effective plans for GUI-related tasks (Lin et al., 2024a). To better understand their performance, we evaluate a setup that isolates their planning ability from grounding. Instead of requiring models to predict both an action and its exact coordinates, we prompt them to generate only a textual description of the action (e.g., "click the file button"). A strong grounding model (UGround-v1) then maps these actions onto the

screen. This approach substantially improves click action recall by 2x to 5x across all models. Specifically, GPT-4o achieves 26.7% recall, Claude reaches 26.9%, and Gemini follows with 25.0%, bringing their performance to near parity. These results indicate that while closed-source models are good at predicting the action, their main limitation is accurately grounding them within the UI.

Open-source GUI agent UI-TARS, achieves the highest overall performance, particularly in click actions with a recall of 19.7%. However, these models often fail to follow instructions accurately, likely due to rigid behavior from training on specific instruction formats. For instance, ShowUI fails to generate drag actions entirely, while UI-TARS occasionally inserts `wait()` and `finish()` actions, even when they are not specified in the prompt. This suggests that while open-source models exhibit stronger grounding, their instruction adherence remains a challenge.

**Error Analysis on Action Prediction.** We identify three major sources of errors in the Action Prediction task (see Appendix D.3). (i) *Poor grounding:* Models often predict the correct action but fail to execute it in the correct location on the screen, leading to grounding errors. (ii) *Lack of platform knowledge:* Some models lack platform-specific knowledge, causing them to hallucinate or predict incorrect actions when they do not fully understand the UI. (iii) **Complexity of the platforms:** Performance tends to be lower on visually dense platforms with many small UI elements, making action prediction more challenging. For example, UI-TARS has the highest error rate (85%) on creativity platforms, where UI elements are tightly packed, whereas education platforms, with simpler layouts, show the lowest error rate (72%).

**More Analysis.** For further insights into factors affecting model performance on all three tasks, we provide additional experiments in Appendix E, including the impact of screenshot resolution (Appendix E.1), cross-software generalization (Appendix E.2), latency trade-offs (Appendix E.3), and an ablation study on planner vs. grounding contributions (Appendix E.4).

## 5. Conclusion

We present UI-Vision, a large-scale GUI benchmark covering 83 desktop applications, making it one of the most extensive and diverse dataset of its kind. We use UI-Vision to build benchmarks supporting three structured evaluation tasks: (i) Element Grounding; (ii) Layout Grounding; and (iii) Action Prediction. Our experiments with state-of-the-art VLMs highlight significant challenges, such as the difficulty in grounding UI elements and predicting actions accurately. The results underscore key gaps in current model capabilities, emphasizing the need for further research in UI interaction modeling for desktops.

## Impact Statement

This research benchmark and analysis offers contributions that are crucial for the advancement of visual-centric UI assistants, specifically targeting agents for Desktop-based software and their applications. The rise of these agents promises to revolutionize how humans engage with digital software and transform the execution of digital tasks by workers. In the following sections, we will discuss both the potential positive and negative societal impacts stemming from this research.

**Positive Impacts:** Enabling GUI agents to automate repetitive and complex desktop tasks allows users to achieve higher productivity and focus on more creative and strategic activities. With more studies and capable GUI agents, software developers can easily finish unit-tests and collect feedback to develop better software or an interface. UI-Vision also provides a standardized desktop benchmark to drive innovation in GUI automation and advance the development of GUI models.

**Potential Negative Impacts:** While our work advances GUI automation, it also presents several challenges. First, protecting user privacy is crucial when developing, evaluating or deploying GUI agents on personal devices, as these systems may inadvertently expose sensitive information. Second, the high computational cost of vision-language models (VLMs) during inference raises significant environmental concerns due to increased energy consumption, which must be addressed for sustainable deployment. Lastly, reliance on highly capable GUI agents may lead to user over-dependence, diminishing manual proficiency, and critical problem-solving skills over time.

## Acknowledgments

Authors would like to sincerely thank Ghazwa Darwiche, Christian Hudon, Tom Murray, Ryan Ghiselli, Pierre-Andre, Chao Wang, and Fanny Rancourt for their valuable administrative and technical support. We also thank Megh Thakkar for his valuable feedback on the paper. Furthermore, we acknowledge MITACS for supporting the recruitment of visiting student researchers who played a significant role in this project.

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

# Appendix

## Table of Contents

| Category | Software Applications |
|---|---|
| **Education** | Anki, Zotero, GrassGIS, Calibre, Audacity, QGIS, OpenBoard, Mendeley |
| **Browsers** | Brave, Chromium, Mozilla Firefox, DuckDuckGo |
| **Development** | VSCode, Atom, FreeCAD, Eclipse, NetBeans, PyCharm, IntelliJ IDEA, Brackets, Geany, Bluefish, KDevelop, Komodo Edit, Code::Blocks, Qt Creator, Arduino IDE, Spyder, Ubuntu Terminal, Conky, Bash, gedit |
| **Productivity** | LibreOffice Calc, LibreOffice Draw, LibreOffice Impress, LibreOffice Writer, draw.io, Joplin, OpenProject, Affine, Zulip, PDFedit, OnlyOffice Calendar, OnlyOffice Document Editor, OnlyOffice Forms, OnlyOffice PDF Forms, OnlyOffice Presentation, OnlyOffice Spreadsheet, Nextcloud, Gnumeric, Simplenote, Cryptomator, WeKan, 7-Zip, GnuCash, Bitwarden, Metabase, Jitsi, Flameshot, Nemo |
| **Creativity** | Blender, GIMP, Inkscape, Krita, darktable, FontForge, MuseScore, Scribus, OpenShot, OBS Studio, Lightworks, Shotcut, Natron, OpenToonz, WordPress |
| **Entertainment** | VLC Media Player, Kodi, Element, Signal, Mastodon, Lemmy, Matrix, Emby |

Table 6: Mapping from coarse-grain categories to platforms. This table lists the categories and their corresponding platforms used in our study.

## A. Human Annotation

We partnered with Turing[4] (referred to as the "Vendor"), a data labeling company that specializes in data curation for AI applications. The annotation process spanned roughly over three three-month periods, with the initial month dedicated to a pilot phase. During this pilot period, we worked closely with the vendor annotation team conducted detailed reviews, and provided extensive feedback to help annotators better grasp the task requirements.

The Vendor annotation team consisted of 70 annotators. This included a multi-tiered team comprising annotators, quality assurance specialists, and managers. The majority of the team was based in India and Latin America. The annotators were in the 20–35 year-old age group, with an equal distribution of male and female participants. They all had strong proficiency in technical writing and English. All of the annotators hold bachelor's degrees in Engineering, Computer Science, and related disciplines. Annotators also had prior experience in data labeling and UI research.

To ensure high-quality annotations, annotators underwent a training process to familiarize themselves with the platforms and the applications. Annotators were paid per hour, and each task took an average of 60 to 90 minutes to complete, from task creation to quality check. The annotators started with creating computer use tasks for 83 different software applications highlighted in Table 6. These computer use tasks were verified by quality assurance specialists and the authors to ensure diversity and coverage. This was followed by the execution of computer use tasks and screen recording. A proprietary tool captured action trajectories and logged the precise coordinates of actions as the annotators performed the task. Finally, the annotators used a custom annotation tool to capture screenshots (keyframes) of each user interaction and refine the action trajectories. Each of the captured keyframes was annotated by drawing bounding boxes around all interface elements visible on the screen. The entire process underwent rigorous quality assurance, including review by human annotators and custom annotation evaluation scripts.

## B. UI-Vision Benchmark Tasks

### B.1. Element Grounding

To create a high-quality benchmark, we implement a systematic data workflow to ensure accuracy, diversity, and reliability for evaluating GUI models. The following steps outline this process:

We start with the bounding box annotations obtained during the initial data collection, as highlighted in Section 3.1. Figure 13 illustrates an example screenshot with annotations. Given the large number of annotations, we sample a subset for the

---

[4]https://www.turing.com/

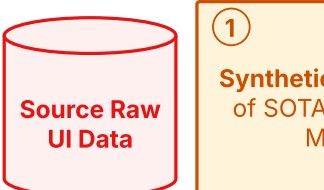 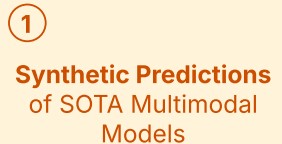 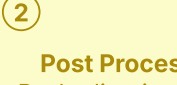 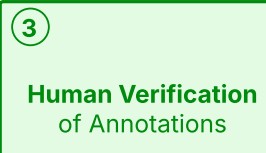 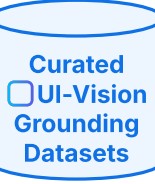

Figure 6: **UI-Vision's Element Grounding Data Curation Workflow.** Step-by-step process from sourcing raw UI data to creating curated datasets. The workflow includes synthetic predictions using state-of-the-art (SOTA) multimodal models, post-processing for deduplication, and human verification to ensure high-quality annotations.

benchmark to ensure comprehensive platform coverage while maintaining ease of evaluation.

Our analysis of state-of-the-art models like UI-TARS (Qin et al., 2025) reveals that they perform well on naive data samples, particularly those with textual information, but struggle significantly with icon-based elements and smaller UI components. Additionally, models show lower performance on platforms with a high bounding box ratio, such as those in the creativity category like Blender and GIMP. To address these issues, we sample bounding boxes from different image resolutions and sizes, with a focus on smaller screen elements.

To maximize diversity, we apply an Intersection over Union (IoU) threshold within each platform, filtering out redundant samples with similar functionality or spatial positioning. This ensures that the selected samples vary in function and placement within the UI. After the initial sampling, we use an ensemble of four top-performing GUI models (e.g., (Qin et al., 2025; Gou et al., 2024b; Yang et al., 2024b)) to identify additional challenging cases. Samples are classified as *hard* (where all models fail or predictions disagree) or *easy* (where at least one model provides a correct prediction), ensuring a balanced difficulty distribution. Reviewers verify uncertain instances by consulting documentation or online sources.

To expand the evaluation beyond basic queries, we introduce variations to assess model capabilities in **functional** and **spatial** settings. This approach allows us to reuse existing annotations while benchmarking different aspects of GUI understanding.

For the **functional** setting, we use each element's annotation and corresponding screenshot (with the bounding box highlighted) as input to GPT-4o to generate function-based descriptions. These descriptions reflect how the element is used rather than just its label.

---

**≣ Prompt used with GPT-4o to generate annotation for Element Grounding with functional setting**

```
You are an assistant tasked with generating a single, precise functional
annotation for a UI element based on its basic textual description.  Your job
is to:
1.  Ensure the functional annotation is accurate, straightforward, and
directly aligns with the provided description and bounding box context before
adding any complexity or diversity.
2.  Focus on clarity and correctness first, avoiding ambiguous or overly
complex annotations.
3.  Use the textual description and bounding box context to infer the most
likely purpose or role of the UI element.
4.  Try best not to directly copy the textual description, but rather provide
a functional interpretation based on the context.

Below is the input for generating a functional annotation:
1.  Screenshot:  Please refer to the provided image.  The UI element is
highlighted by a red bounding box.
2.  Description of the UI element (basic textual content):  "{annotation for
basic setting}".
```

```
Ensure the annotation meets the following requirements:
- It directly reflects the UI element's functionality as described in the
textual content.
- It is clear, concise, and free of unnecessary complexity.
- It avoids speculative or ambiguous statements.
- It clearly reflects how the user interacts with the UI element.

Provide only the JSON output without additional explanations.
```

For the **spatial** setting, we iterate over four directions—[up, bottom, left, right]—to generate queries about the closest neighboring element. For example, given an element labeled "previous annotation options", a corresponding query might be: "What is the element that is immediately to the right of 'previous annotation options' horizontally?" We retrieve the correct bounding box for the queried element using the bounding boxes collected during initial data creation (Section 3.1). To ensure accuracy and reduce ambiguity, we remove samples where no valid closest element exists or where the distance exceeds a set threshold. The processing scripts will be released alongside the code for reproducibility.

### B.2. Layout Grounding

GUI layouts define how interface components, such as buttons, text fields, and containers, are arranged into cohesive regions (e.g., "Formatting Tools" or "Main Navigation Bar"). In our layout grounding task, we start with raw UI element annotations collected during the initial data collection phase, which include dense bounding box information for individual elements. These annotations are then fed into LLAMA-3.3-70B (Meta, 2024) using a predefined prompt (provided below) that instructs the model to cluster UI elements into non-overlapping functional and semantic groups. For each group, the model generates a corresponding textual description and computes an aggregated bounding box that encompasses all the elements within that group, thereby capturing the higher-level structure of the GUI.

Following the automated model inference, the predicted clusters and bounding boxes are subjected to a rigorous human verification process. Expert annotators in our research team review and validate each functional group to ensure that the clusters accurately represent the intended UI regions and that the aggregated bounding boxes fully cover the grouped elements. Specifically, we remove all samples that do not meet the requirements and retain those that do. This two-stage procedure yields a high-quality dataset of 311 human-verified query-label pairs spanning 77 platforms, with each functional group typically containing 5–10 UI elements. The process thus extends individual element grounding to a comprehensive layout grounding task, emphasizing both spatial organization and semantic coherence in desktop interfaces. To the best of our knowledge, this is the first work to introduce this setting and systematically evaluate GUI agents in this context.

**Prompt used with LLAMA-3.3-70B to generate samples for layout grounding**

```
Below is a list of dictionaries representing UI elements for the software
'{software_name}'. Please use the knowledge you have on this software to
analyze these elements and group them based on their functionality.

Your task is to:

1.  Analyze each element for text, category, and boundingBox attribute.
2.  Identify the elements that belong to a particular functional group (e.g.,
all the file edit tools). Name as many as you possibly can, but make sure the
functional groups make sense.
3.  Combine the bounding boxes of all elements belonging to that functional
group into a single bounding box that encompasses them all.
4.  Return a structured response (JSON or similarly structured text) that
includes:  - The name of the group (e.g., "Edit Tools Region").
- An explanation of the group's functionality.  Please be as specific as
```

```
possible but do not simply repeat the names of the UI elements in the group.
- The coarse bounding box (x1, y1, x2, y2, etc.).

Here is the list of dictionaries you should work with:
{raw data for the given screenshot}

Please produce your final answer in a structured JSON-like format, such as:

[
    {
        "name": "Edit Tools Region",\\[1ex]
        "explanation": "This region contains all the tools related to
                        editing the files.",
        "boundingBox": \{
            "x1": ...,
            "y1": ...,
            "x2": ...,
            "y2": ...
        \}
    \},
    ...
]

Some additional requirements:

1.  Please try your best to make functional groups that cover all the elements
in the list.
2.  Please make the functional group not overlap with each other.  Try to
prioritize non-overlapping bounding boxes containing more elements.
3.  Make sure the name of each group is descriptive as well as meaningful to
the human user and relevant to the elements it contains.  Please make sure the
name is not too generic.
4.  Make the bounding box complete, based on the bounding boxes of the
elements it contains.

Make sure to provide the accurate bounding box that encloses all relevant
elements for each functional group.  And return only this JSON object.
```

### B.3. Action Prediction

We begin with the raw action trajectories collected and annotated by humans during the initial data collection stage. Table 7 lists all recorded actions. Our analysis revealed that current GUI models are limited in their ability to predict a wide range of actions, as they are typically trained on a restricted set. To simplify evaluation, we standardized actions into five categories: click, move to, drag, typing, and hotkey. These actions are common across all models. We excluded tasks involving key down, key up, and scroll actions, reducing the total number of tasks to 442. Additionally, we applied the following preprocessing steps to refine the action data:

1. **Removing Redundant Move Actions:** If a "move to" action directly preceded a "click" action, we removed it. Since the "click" action already contains positional data, the preceding movement was unnecessary.

2. **Grouping Drag Actions:** A drag action is always preceded by a mouse-down event and followed by a mouse-up event. We merged these three actions into a single "drag" action. Additionally, since a drag operation requires moving the cursor to the starting position, we included this movement, resulting in a standardized format: drag(x1, y1, x2, y2) where x1, y1 is the start of the drag and x2, y2 is the end coordinates of the drag.

3. **Merging Press and Hotkey Actions:** In the raw trajectories, a "press" action referred to pressing a single key, whereas

| Action Type | Parameters | Description |
|---|---|---|
| MOVE_TO | *x, y* | Move the cursor to the specified position |
| CLICK | *button,* | Click the mouse button at a specified location |
|  | *x, y,* |  |
|  | *num_clicks* |  |
| MOUSE_DOWN | *button* | Press and hold the specified mouse button |
| MOUSE_UP | *button* | Release the specified mouse button |
| DRAG_TO | *x, y* | Drag the cursor to the specified position with the left button pressed |
| SCROLL | *dx, dy* | Scroll the mouse wheel up or down |
| TYPING | *text* | Type the specified text |
| KEY_DOWN | *key* | Press and hold the specified key |
| KEY_UP | *key* | Release the specified key |
| HOTKEY | *key/keys* | Press the specified key or key combination |

Table 7: Table of original Mouse and Keyboard Actions during human demonstration.

a "hotkey" involved multiple keys. We merged both into a single "hotkey" category to maintain consistency.

## C. Data Statistics and Examples

**Data Statistics.** Figure.12 presents key statistical distributions within the UI-Vision. (a) Shows the distribution of bounding box number per screenshot, indicating a mean of 74.3, reflecting the dense UI elements present in each frame. (b) Illustrates the distribution of human recording durations in UI-Vision, with a mean of 38.2 seconds, highlighting the variation in the GUI navigation task. (c) Depicts the distribution of the number of action steps required to complete tasks, with a mean of 14.0, indicating the long-horizon challenges and interaction complexity within the dataset. (d) Shows the distribution of different screenshot resolutions in our dataset. We have more than 100 screenshot resolutions offering rich diversity for benchmarking.

**Visualization.** Figure 7 to 11 illustrate concrete examples highlighting different aspects and challenges of UI-Vision's tasks across multiple software applications. In Figure 7, we demonstrate examples from the Element Grounding task with the basic setting, showing typical UI elements identified by straightforward textual descriptions. Models must accurately localize UI components based solely on labels like "crop tool" or "skip all breakpoints". Figure 8 presents cases from the Element Grounding task with the functional setting, emphasizing the need for models to understand UI elements based on their functionality rather than simple visual labels, such as "Remove data series from the chart" or "Open the DoxyBlocks plugin menu". Figure 9 provides challenging examples from the Element Grounding task with spatial setting, requiring models to precisely interpret spatial relations among UI components, such as elements directly adjacent or positioned relative to specific reference elements. Lastly, Figure 10 demonstrates examples from Layout Grounding, showcasing how models must group UI components into meaningful semantic regions, like identifying comprehensive areas for menu options, editing tools, or formatting tool clusters. Figure 11 shows an example of Action Prediction task along with the task and raw actions collected.

## D. Error Analysis

### D.1. Element Grounding

Below, we present several failure cases predicted by the top-performing UI-TARS model (Qin et al., 2025) to examine the limitations of current models. We highlight the ground truth with a red box. The prediction of the model is indicated by a red arrow surrounding it.

**A. Fine-grained ambiguity:** As shown in Fig. 14, the agent effectively understands the query and primarily responds to the

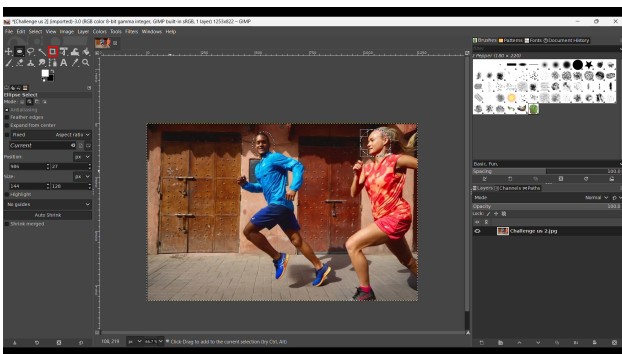

(a) Query: "crop tool". The ground truth element is bounded by red bounding box. The platform of the example is **GIMP**.

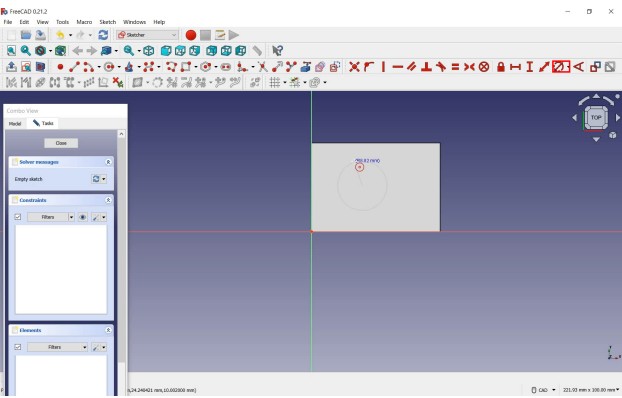

(b) Query: "constrain arc or circle". The ground truth element is bounded by red bounding box. The platform of the example is **FreeCAD**.

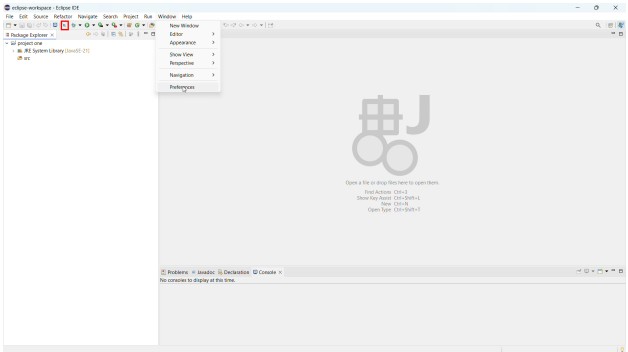

(c) Query: "skip all breakpoints". The ground truth element is bounded by red bounding box. The platform of the example is **Eclipse**.

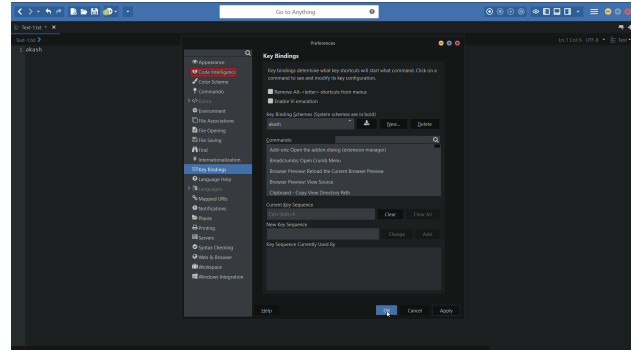

(d) Query: "code intelligence". The ground truth element is bounded by red bounding box. The platform of the example is **Komodo Edit**.

Figure 7: Examples of Element Grounding with basic setting. The ground truth element is bounded by red bounding box. We demonstrate examples from the Element Grounding task with the basic setting, showing typical UI elements identified by straightforward textual descriptions. Models must accurately localize UI components based solely on labels like "crop tool" or "skip all breakpoints".

nearby region but struggles to recognize the correct target among several similar candidates. This highlights the need for existing agents to develop advanced validation strategies.

**B. Lack of domain knowledge :** In Fig.15, we observe that the model's difficulty in identifying the most efficient approach to task completion underscores a deficiency in domain-specific knowledge, such as understanding what the letter "F" signifies. This challenge prompts us to consider developing solutions based on Retrieval-Augmented Generation (RAG) or integrating external databases, such as software documentation or tutorials, to enhance the model's performance.

**C. Small elements:** In Fig.16, we find that the model struggles with small visual elements, especially in an interface with high resolution, and more importantly, a dense distribution of UI elements. To resolve this issue, one potential strategy is to develop an iterative zoom-in strategy as V-Search (Wu & Xie, 2024).

**D. Cross-platform generalization:** The model sometimes incorrectly transfers layout assumptions across platform. As in Fig.17, the query given to the model is minimize window. The model fails to generalize to the "minimize" button for IOS system and points to the place where "minimize" button is located in the Windows operating system. We highlight the ground truth with a red box. The prediction of the model is indicated by a red arrow surrounding it.

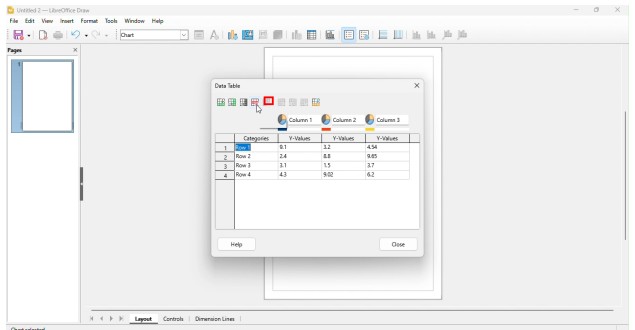

(a) Query: "Remove data series from the chart". The ground truth element is bounded by red bounding box. The platform of the example is **LibreOffice Draw**.

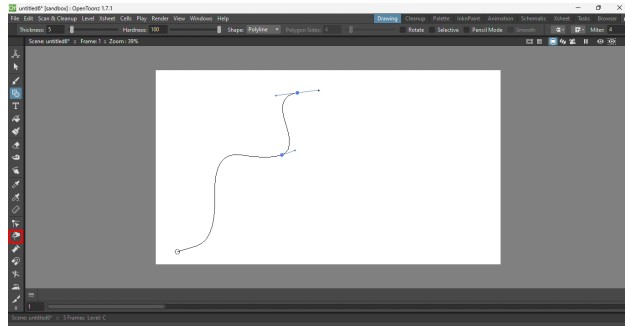

(b) Query: "Create a new Java package". The ground truth element is bounded by red bounding box. The platform of the example is **Eclipse**.

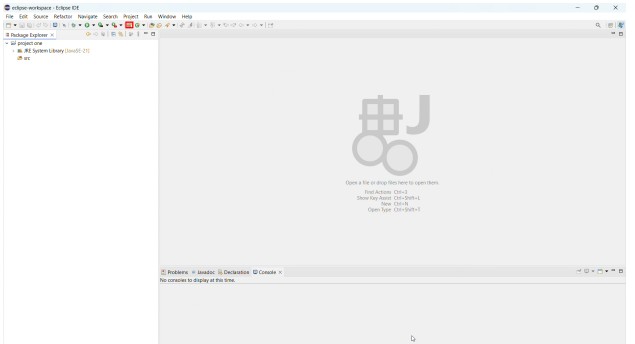

(c) Query: "Adjusts path curvature by pinching vector points". The ground truth element is bounded by red bounding box. The platform of the example is **OpenToonz**.

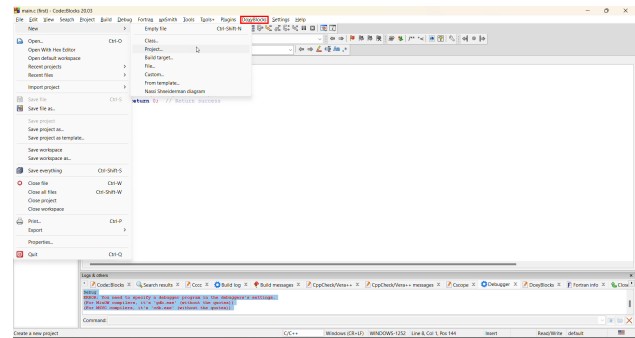

(d) Query: "Open the DoxyBlocks plugin menu". The ground truth element is bounded by red bounding box. The platform of the example is **Code::Blocks**.

Figure 8: Examples of Element Grounding with functional setting. The ground truth element is bounded by red bounding box. We present cases from the Element Grounding task with the functional setting, emphasizing the need for models to understand UI elements based on their functionality rather than simple visual labels, such as "Remove data series from the chart" or "Open the DoxyBlocks plugin menu".

### D.2. Layout Grounding

Below, we present some failure cases predicted by the top-performing closed-source model Gemini-1.5-pro (Team et al., 2024) and open-source GUI agent OSAtlas-7B (Wu et al., 2024) to examine the limitations of both types of models as their error patterns are not identical. We highlight the prediction of the model with a blue box. The ground truth is indicated by a red arrow surrounding it.

**A. Inaccurate bounding box placement:** As shown in Figure 18a and Figure 19a, it is quite typical for closed-source models to fail to return the minimal bounding box of the ground truth region, though it is usually contained in the predicted box. This indicates models can not understand the partition of the layout well.

**B. Poor functional grouping:** As shown in Figure 18b, open-source GUI agents sometimes struggle even with a very specific query that explicitly mentions the names of almost all the elements. Usually it should not be very difficult for the model to ground those elements one at a time in the basic Element Grounding setting. This indicates quite a weak understanding of functional groups with coarser granularity. Instead, they may rely solely on learning the correspondence between an element's label and its position. This limitation is likely due to the lack of training data that enables models to perceive interfaces at a higher and more generalized level.

**C. Superficial semantic matching:** As shown in Figure 19b, when an open-source agent lacks sufficient knowledge or confidence in locating the requested region, it sometimes predicts only one or two items whose labels share words with the query but are not semantically equivalent. In the example, the predicted item labeled "Design" appears in the query but is an

incorrect selection. This suggests that open-source GUI agents require a deeper semantic understanding of interface layouts beyond simple keyword matching and grounding.

### D.3. Action Prediction

We analyze UI-TARS (Qin et al., 2025), the best-performing open-source model, and Gemini 1.5-Pro (Team et al., 2024), the highest-performing closed-source model, to identify common failure cases in Action Prediction prediction. Our analysis highlights three major sources of errors that impact model performance:

**A. Poor Grounding:** As discussed in Section 4.3, both open-source and closed-source models struggle with accurately grounding actions. In many cases, models correctly predict the intended action but fail to execute it on the correct UI element. Figure 20 illustrates an example where Gemini-1.5-Pro identifies the right action but applies it to the wrong location, leading to failed execution.

**B. Lack of Platform Knowledge:** Each platform has unique UI elements and interactions. To perform tasks effectively, models need an understanding of platform-specific keywords and the major operations each platform supports. Our analysis shows that models often lack this knowledge, likely due to insufficient exposure to diverse desktop environments during training. As a result, models sometimes hallucinate actions or generate nonsensical predictions. Figures 21 and 22 showcase cases where the models' limited platform knowledge leads to incorrect or implausible actions.

**C. Complexity of the Platforms:** Models struggle more with platforms that have dense and intricate UI layouts. Smaller UI elements and tightly packed interfaces make it harder for models to perceive and interact accurately. Additionally, platforms with extensive functionalities allow users to perform complex workflows, increasing the difficulty of predicting appropriate actions. This challenge is reflected in UI-TARS' highest error rate (85%) on creativity platforms, where elements are densely packed, while education platforms, with simpler layouts, exhibit the lowest error rate (72%).

## E. More Experimental Analysis

### E.1. Effect of Screenshot Resolution on Element Grounding Accuracy

Figure 23 presents the grounding accuracy of the two best-performing models, **UI-TARS-72B** and **UGround-v1-72B**, across different screenshot areas (i.e., resolutions). Observing the accuracy distributions, there is no clear trend that indicates a strong correlation between the resolution of the screen and the grounding accuracy. The accuracy values fluctuate across different resolution levels for both models, suggesting that factors other than resolution may have a more significant impact on performance.

### E.2. Cross-software Generalization Analysis

We compare model performance on common apps (e.g., VSCode) vs. less common ones (e.g., FreeCAD, QGIS) using the element grounding subset in R1. Results are included in Table 8. While we cannot confirm the exact training data used in several models, this serves as a proxy for generalization analysis. We observe that all models show significant accuracy drops on less common apps, confirming consistent generalization challenges.

### E.3. Latency Analysis

We report latency per query, average output tokens, and GPU usage across all three tasks using default Hugging Face implementations for consistency. Table 9, 10 and 11 for Element Grounding,Layout Grounding and Action Prediction, respectively. Token efficiency was measured with GPT-4 tokenization. Models like UI-TARS, trained on action-heavy tasks, generate longer outputs due to detailed step-by-step reasoning.

### E.4. Analysis of Planner vs. Grounding Model Contributions

Our experiments demonstrate a significant improvement in recall when large language model planners are combined with grounding models, a gain we attribute primarily to better action selection since the grounding model only supplies coordinates. We further analyzed error rate reductions across platform categories (Fig. 24), finding that productivity tools experienced the largest decrease (26%) despite entertainment tools having a higher baseline grounding accuracy—underscoring the planner's impact—whereas creativity platforms saw the smallest gain (14%), highlighting challenges in planning and grounding

| Model | Basic | | | | Functional | | | | Spatial | | | |
|---|---|---|---|---|---|---|---|---|---|---|---|---|
| | Acc (Common) (205) | Acc (Rare) (173) | Delta (%) | Overall (1772) | Acc (Common) (205) | Acc (Rare) (173) | Delta (%) | Overall (1772) | Acc (Common) (289) | Acc (Rare) (156) | Delta (%) | Overall (1935) |
| *Closed-Source Models* | | | | | | | | | | | | |
| Claude-3.5-Sonnet | 5.85 | 0.58 | 90.09 | 5.08 | 4.39 | 0.00 | 100.00 | 5.19 | 1.73 | 2.56 | -47.98 | 3.15 |
| Claude-3.7-Sonnet | 9.26 | 3.47 | 62.53 | 9.48 | 8.78 | 1.73 | 80.30 | 7.73 | 6.92 | 1.92 | 72.25 | 7.60 |
| *Open-Source VLMs* | | | | | | | | | | | | |
| Qwen2VL-7B | 3.41 | 0.58 | 82.99 | 3.44 | 2.44 | 0.00 | 100.00 | 3.22 | 0.69 | 0.64 | 7.25 | 1.45 |
| MiniCPM-V-8B | 6.34 | 4.62 | 27.13 | 7.11 | 3.41 | 2.31 | 32.26 | 5.30 | 0.35 | 0.00 | 100.00 | 1.45 |
| *Open-Source GUI Agents* | | | | | | | | | | | | |
| ShowUI-2B | 7.80 | 5.20 | 33.33 | 8.07 | 8.78 | 5.20 | 40.77 | 7.67 | 2.08 | 1.28 | 38.46 | 2.07 |
| AriaUI-25.3B | 13.70 | 9.82 | 28.32 | 12.20 | 13.20 | 8.09 | 38.71 | 14.00 | 4.50 | 3.20 | 28.89 | 3.98 |
| OSAtlas-7B | 11.70 | 6.93 | 40.77 | 12.20 | 9.76 | 8.09 | 17.11 | 11.20 | 3.11 | 3.85 | -23.79 | 3.67 |
| UGround-v1-7B | 15.10 | 10.40 | 31.13 | 15.40 | 16.10 | 13.30 | 17.39 | 17.10 | 6.23 | 5.13 | 17.66 | 6.25 |
| Aguvis-7B | 15.60 | 10.40 | 33.33 | 17.80 | 17.10 | 10.40 | 39.18 | 18.30 | 3.46 | 3.21 | 7.23 | 5.06 |
| UI-TARS-7B | 20.50 | 11.60 | 43.41 | 20.10 | 23.40 | 15.60 | 33.33 | 24.30 | 8.30 | 5.13 | 38.19 | 8.37 |
| CogAgent24-9B | 19.50 | 6.36 | 67.38 | 12.00 | 16.10 | 4.05 | 74.84 | 12.20 | 6.23 | 0.64 | 89.73 | 2.63 |
| UGround-v1-72B | 26.80 | 17.90 | 33.21 | 27.90 | 26.80 | 14.50 | 45.90 | 26.70 | 15.90 | 12.80 | 19.50 | 14.90 |
| UI-TARS-72B | 34.60 | 20.80 | 39.88 | 31.40 | 33.70 | 18.50 | 45.10 | 30.50 | 16.60 | 11.50 | 30.72 | 14.70 |

Table 8: **Cross-software generalization analysis for Element Grounding**. Since directly retraining models for evaluation is beyond the scope of this study, we approximate cross-software generalization by comparing model performance on *common apps* (*e.g.*, VSCode, draw.io, OnlyOffice suite) versus *rare apps* (*e.g.*, FreeCAD, QGIS, Qt Creator, darktable, Mastodon), the latter presumably unseen during training. We report accuracy scores on these subsets along with the relative accuracy degradation (delta percentage). Positive deltas indicate better performance on common apps, reflecting a drop when generalized to rare apps. We also add the overall results in Table 3 as a reference. From the results, all models show significant accuracy drops on less common apps, confirming consistent generalization challenge. Notably, larger and extensively trained models (*e.g.*, UI-TARS-72B) exhibit higher overall accuracy but still encounter noticeable performance reductions on rare apps, highlighting the challenges of generalization in practical scenarios. Models are categorized by size for clarity: gray (closed-source models), green (open-source VLMs), and blue / orange for open-source GUI Agents below/above 8B (active) parameters, respectively.

| Model | Avg Latency per Query (s) | Avg # Output Tokens | # GPUs (H100) |
|---|---|---|---|
| *Closed-Source Models (API-based)* | | | |
| Claude-3.5-Sonnet | - | 8.28 | - |
| Claude-3.7-Sonnet | - | 47.5 | - |
| *Open-Source VLMs* | | | |
| Qwen2VL-7B | 1.36 | 116.1 | 1 |
| MiniCPM-V-8B | 0.84 | 28.6 | 1 |
| *Open-Source GUI Agents* | | | |
| ShowUI-2B | 4.73 | 25.9 | 1 |
| AriaUI-25.3B | 2.38 | 14.0 | 4 |
| UI-TARS-7B | 4.97 | 66.1 | 2 |
| Aguvis-7B | 1.38 | 18.3 | 1 |
| OSAtlas-7B | 0.76 | 40.6 | 1 |
| UGround-7B | 0.60 | 6.42 | 1 |
| CogAgent24-9B | 6.14 | 173.2 | 1 |
| UI-TARS-72B | 12.1 | 57.6 | 4 |

Table 9: Efficiency metrics for the Element Grounding task under the **Basic** setting. Latency measurements are based on default Hugging Face implementations via the Transformers library, ensuring fair comparison (not applicable to closed-source models). Output tokens are counted using the GPT-4 tokenizer. Models are grouped into three categories: gray for closed-source models (API-based), green for open-source VLMs, and blue / orange for open-source GUI Agents below/above 8B (active) parameters, respectively.

| Model | Avg Latency per Query (s) | Avg # Output Tokens | # GPUs (H100) |
|---|---|---|---|
| *Closed-Source Models (API-based)* | | | |
| GPT-4o | - | 33.9 | - |
| Claude-3.5-Sonnet | - | 166.2 | - |
| Gemini-1.5-pro | - | 20.1 | - |
| *Open-Source VLMs* | | | |
| MiniCPM-V-8B | 0.66 | 18.8 | 1 |
| Qwen2VL-7B | 0.95 | 25.7 | 1 |
| *Open-Source GUI Agents* | | | |
| OSAtlas-7B | 0.95 | 39.8 | 1 |
| CogAgent24-9B | 6.71 | 192.1 | 1 |

Table 10: Efficiency metrics for the Layout Grounding task. Latency measurements are based on default Hugging Face implementations via the Transformers library, providing a fair comparison across models (not applicable to closed-source models). Output tokens are counted using the GPT-4 tokenizer. Models are grouped into three categories: gray for closed-source models (API-based), green for open-source VLMs, and blue / orange for open-source GUI Agents below/above 8B parameters, respectively.

| Model | Avg Latency per Query (s) | Avg # Output Tokens | # GPUs (H100) |
|-------|--------------------------|---------------------|---------------|
| *Closed-Source Models (API-based)* | | | |
| GPT-4o | - | 59.6 | - |
| Claude-3.5-Sonnet | - | 65.3 | - |
| Gemini-1.5-pro | - | 55.9 | - |
| *Open-Source GUI Agents* | | | |
| ShowUI-2B | 0.6 | 25.3 | 1 |
| UI-TARS-7B | 1.7 | 58.0 | 1 |

Table 11: Efficiency metrics for the Action Prediction task. Latency measurements are based on default Hugging Face implementations via the Transformers library, providing a fair comparison across models (not applicable to closed-source models). Output tokens are counted using the GPT-4 tokenizer. Models are grouped into two categories: gray for closed-source models (API-based), and blue for open-source GUI Agents. We use FlashAttention-2 in our implementation for all open-source models.

within functionally dense interfaces featuring small UI elements.

# F. Limitation and Future works

Our benchmark tasks evaluate models in an offline setting. However, expanding it to an online environment would enable a more comprehensive assessment of real-world interactions. We currently use heuristic sampling methods to select challenging UI elements for the benchmark. Future work should explore improved strategies to ensure more representative UI elements are sampled for evaluating agentic abilities. In the action task, we rely on a single human demonstration, which may not fully capture the range of possible user interactions. Since tasks can often be completed in multiple ways, a single demonstration might not provide a complete assessment of model capabilities. Expanding the dataset to include multiple action trajectories per task would offer a more robust evaluation and reduce potential biases. Additionally, we have not yet incorporated human-recorded videos, which are crucial for assessing GUI action understanding. Providing video clips and querying models about possible actions could offer deeper insights into their reasoning and decision-making processes. Finally, our benchmark does not explicitly evaluate combined actions involving both mouse and keyboard inputs, such as holding the Control key while dragging an item. Since most models have limited exposure to such interactions, developing reliable evaluation methods for these complex actions remains an open challenge.

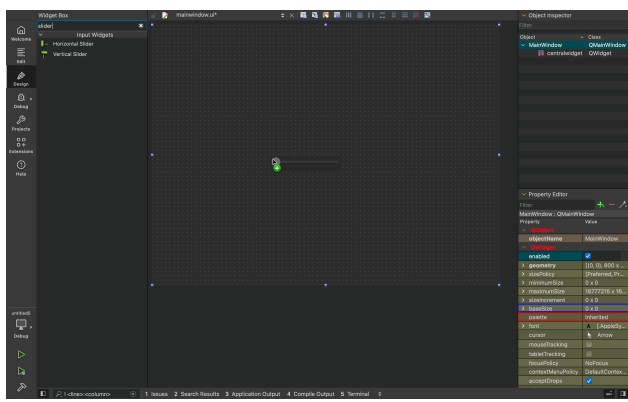

(a) Query: "Directly vertically below the `"baseSize",` can you find the closest element?". The element in the blue box is the reference one referring to "medium eraser" while the red one is the ground truth. The platform of the example is **Qt Creator**.

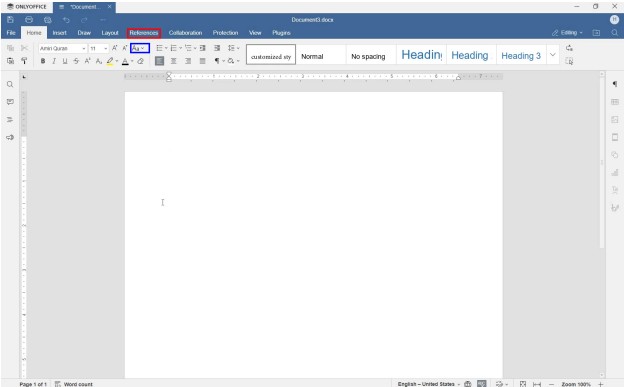

(b) Query: "What is the element that is vertically closest to the `"change case"` and above it?". The element in the blue box is the reference one referring to "medium eraser" while the red one is the ground truth. The platform of the example is **OnlyOffice Document Editor**.

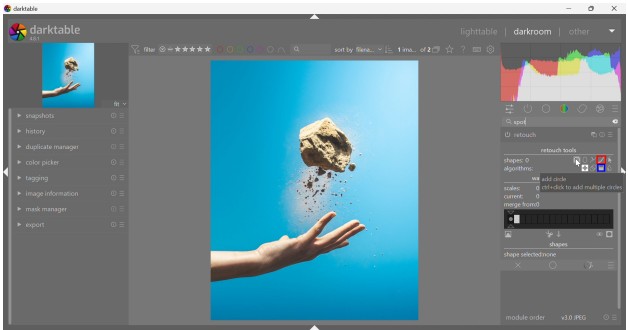

(c) Query: "What is the element that is vertically closest to the `"Activate fill tool"` and above it?". The element in the blue box is the reference one referring to "medium eraser" while the red one is the ground truth. The platform of the example is **Darktable**.

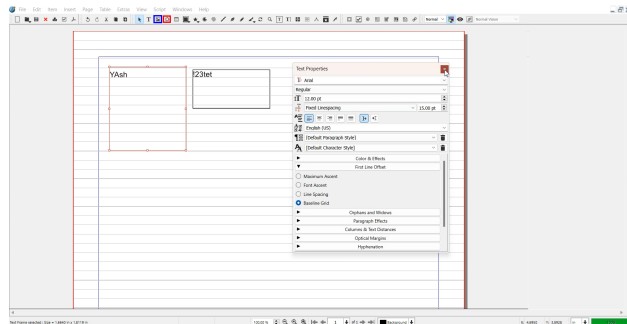

(d) Query: "What is the element that is horizontally closest to the `"Image Frame"` and to the right of it". The element in the blue box is the reference one referring to "medium eraser" while the red one is the ground truth. The platform of the example is **Scribus**.

Figure 9: Examples of Element Grounding with spatial setting. The ground truth element is bounded by red bounding box. We provide challenging examples from the Element Grounding task with spatial setting, requiring models to precisely interpret spatial relations among UI components, such as elements directly adjacent or positioned relative to specific reference elements.

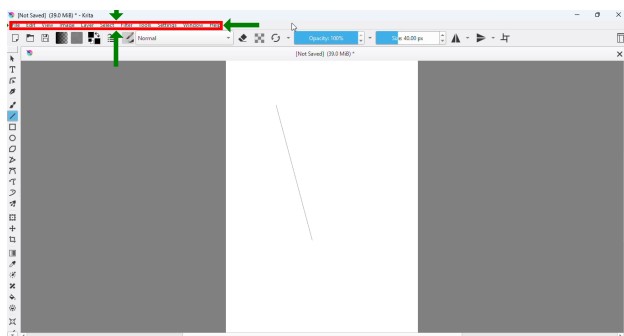

(a) Query: "Menu Bar: this region contains the main menu options for the application, including file, edit, view, and more." The red is the ground truth. The platform of the example is **Krita**.

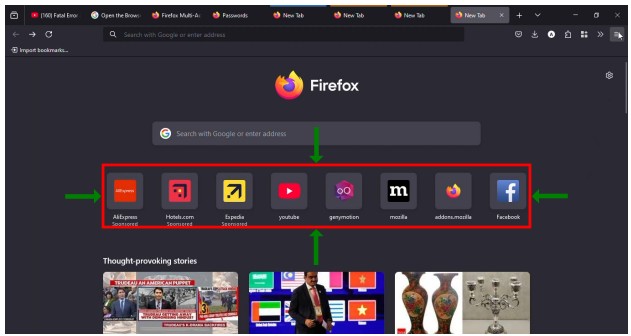

(b) Query: "Shortcut Region: this region contains shortcuts to various actions and features such as bookmarks, history, and settings." The red is the ground truth. The platform of the example is **Mozilla Firefox**.

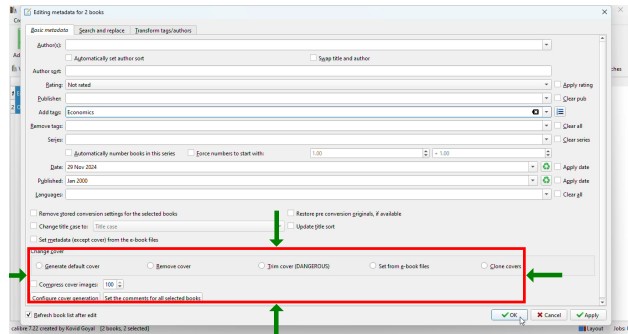

(c) Query: "Cover Management Region: this region contains all the tools and options related to managing book covers, such as changing, removing, and generating covers." The red is the ground truth. The platform of the example is **Calibre**.

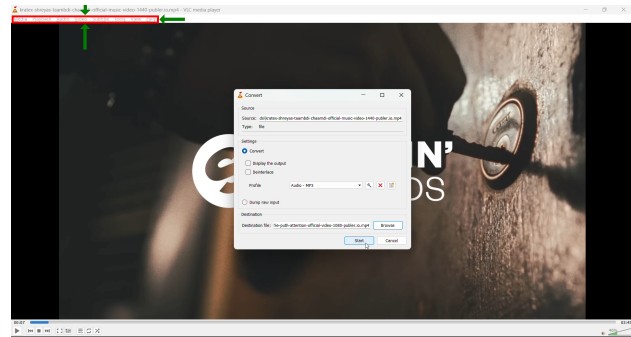

(d) Query: "Top Menu Bar: this region contains the main menu options for the application, including Media, Playback, Audio, Video, and Help." The red is the ground truth. The platform of the example is **VLC Media Player**.

Figure 10: Examples of Layout Grounding. The ground truth element is bounded by red bounding box. We demonstrate examples from Layout Grounding, showcasing how models must group UI components into meaningful semantic regions, like identifying comprehensive areas for menu options, editing tools, or formatting tool clusters.

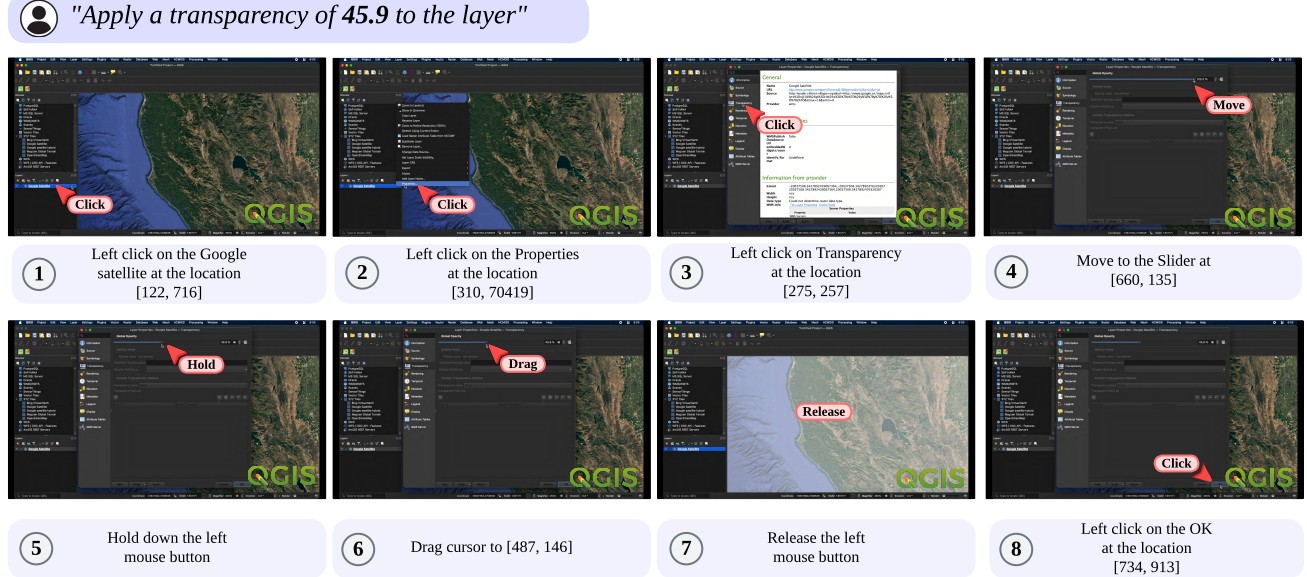

Figure 11: The figure illustrates the raw action trajectories recorded for a computer use task in QGIS software. To simplify evaluation, certain actions, such as drag, are grouped with related actions like mouse up and mouse down, while redundant actions, such as move, are removed. The recorded data consists only of action coordinates, which have been translated into descriptive terms for clarity.

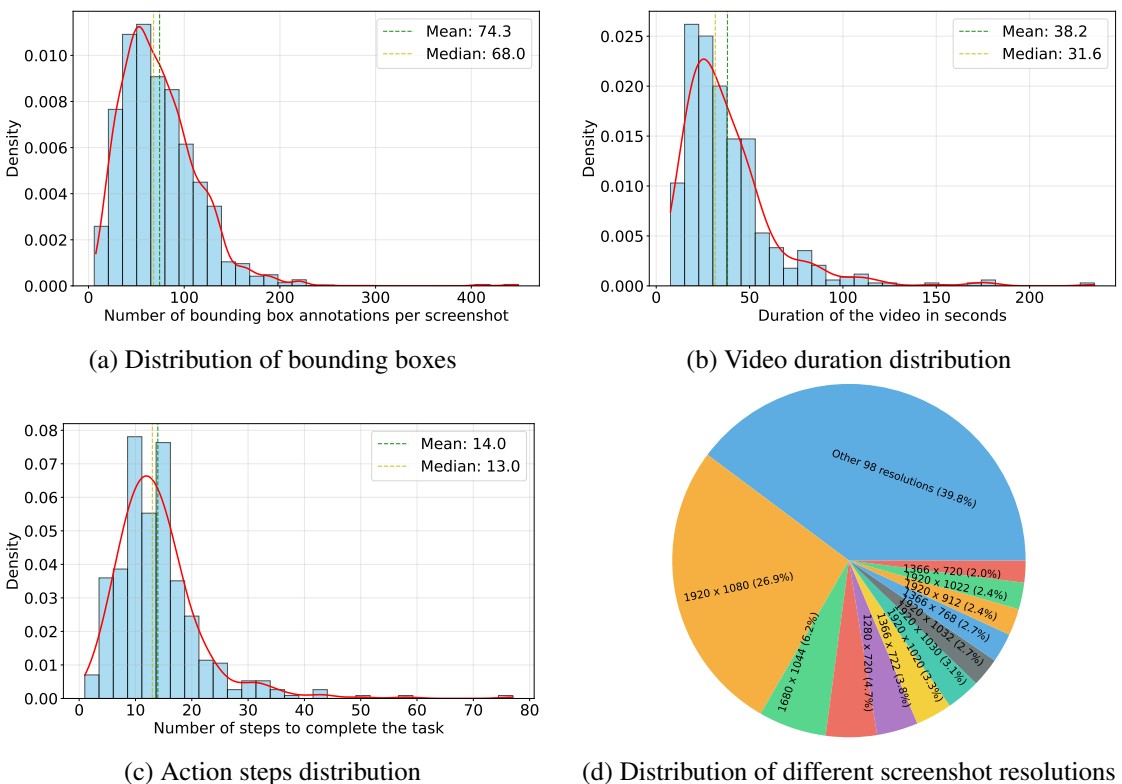

(a) Distribution of bounding boxes

(b) Video duration distribution

(c) Action steps distribution

(d) Distribution of different screenshot resolutions

Figure 12: **Dataset Statistics.** (a) Distribution of the number of bounding boxes present per keyframe. (b) Distribution of video durations in UI-Vision. (c) Distribution of the number of actions required to complete the task.

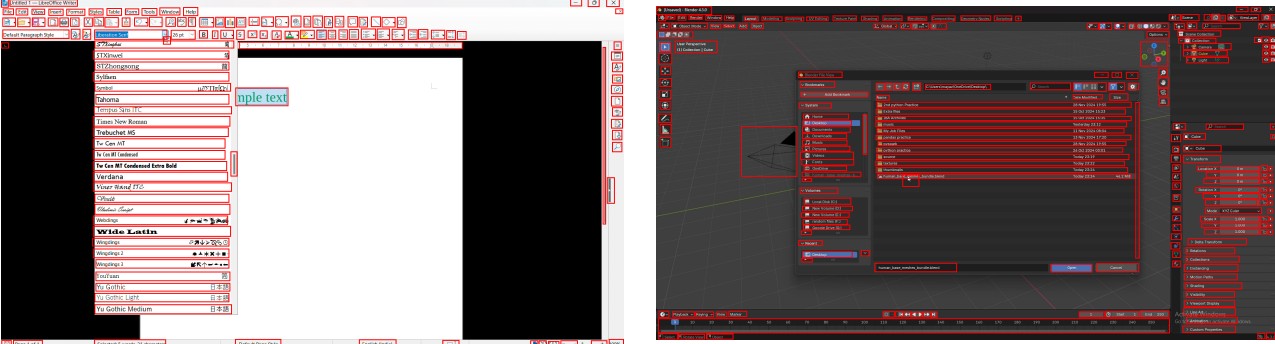

Figure 13: Examples of raw annotation of a screenshot in UI-Vision. It is a representative of dense and complete element coverage of bounding box annotations.

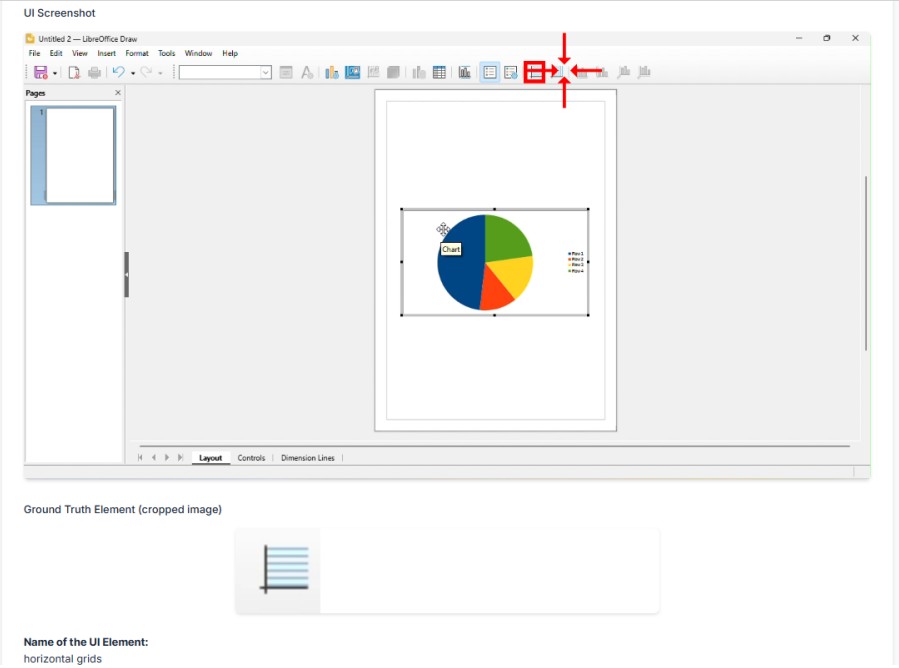

Figure 14: Typical error case (A) of Element Grounding with basic setting. The query given to the model is `horizontal grids`. The model fails to identify the fine-grained differences in those cases. We highlight the ground truth with a red box. The prediction of the model is indicated by a red arrow surrounding it. The ground truth element is cropped and attached below the screenshot of each interface.

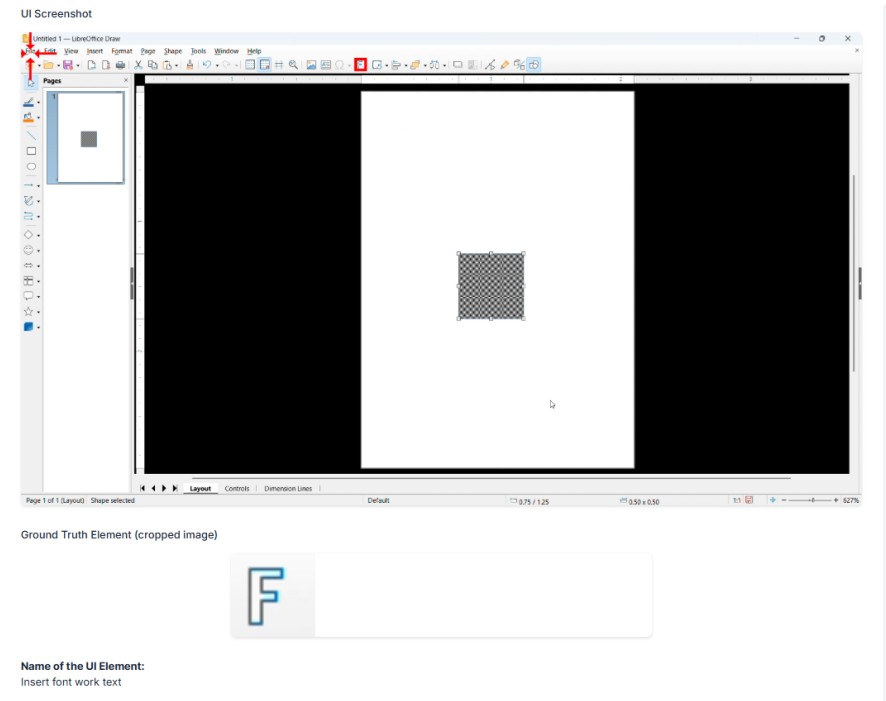

Figure 15: Typical error case (B) of Element Grounding with basic setting. The query given to the model is `insert font work text`. The model shows the obvious lack of domain knowledge of the software/platform in question. We highlight the ground truth with a red box. The prediction of the model is indicated by a red arrow surrounding it. The ground truth element is cropped and attached below the screenshot of each interface.

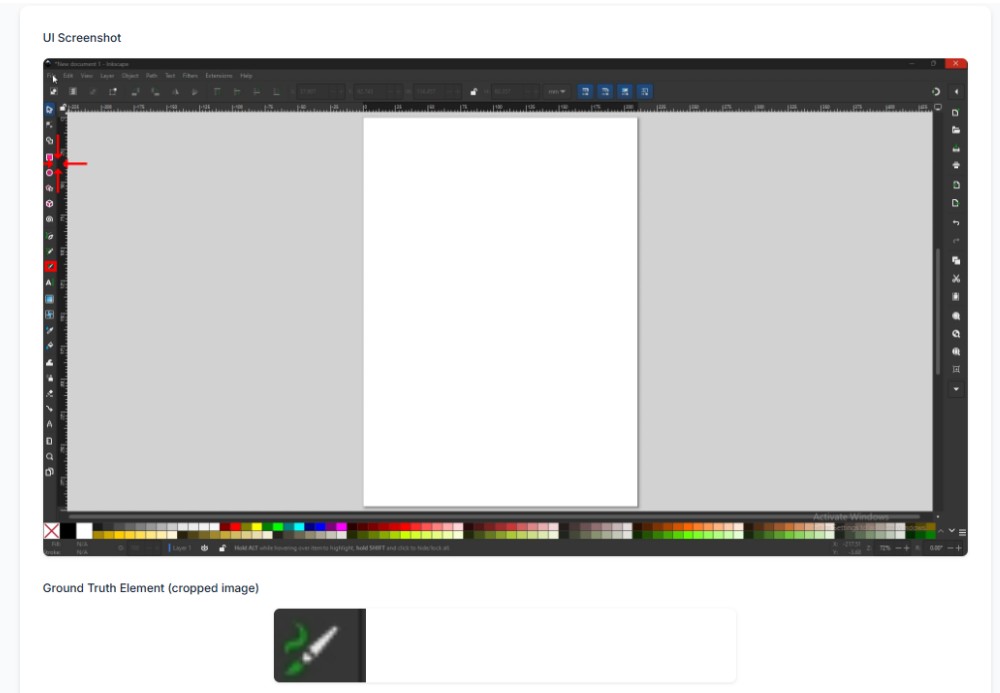

Figure 16: Typical error case (C) of Element Grounding with basic setting. The query given to the model is `Calligraphy`. The model fails to locate and understand small items within an interface with a dense distribution of UI elements. We highlight the ground truth with a red box. The prediction of the model is indicated by a red arrow surrounding it. The ground truth element is cropped and attached below the screenshot of each interface.

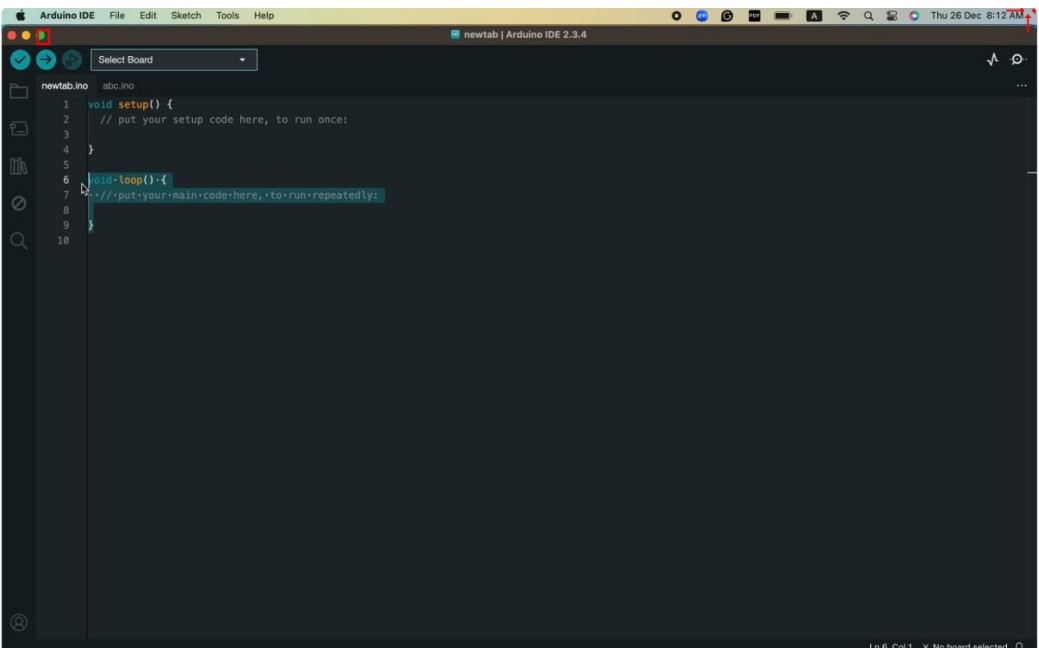

Figure 17: Typical error case (D) of Element Grounding with basic setting. The query given to the model is `minimize window`. The model fails to generalize to the "minimize" button for IOS system and points to the place where the "minimize" button is located in the Windows operating system. We highlight the ground truth with a red box. The prediction of the model is indicated by a red arrow surrounding it.

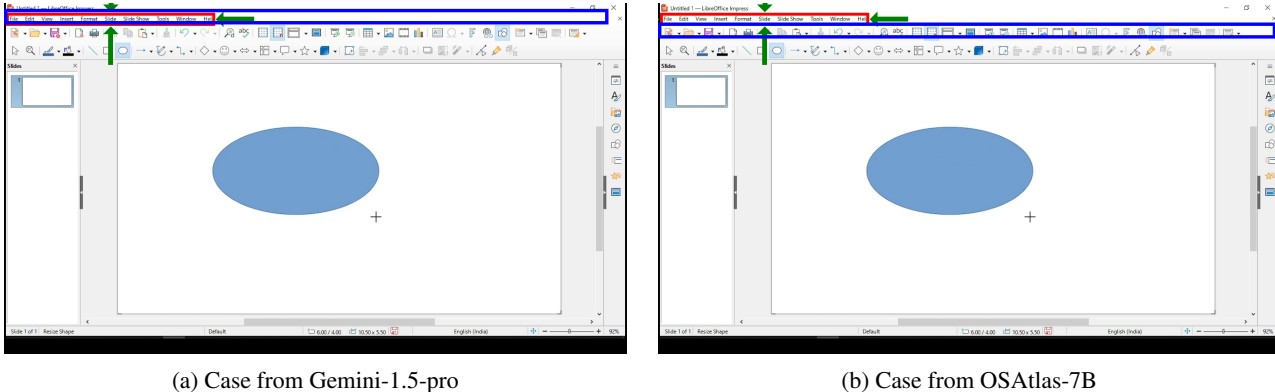

(a) Case from Gemini-1.5-pro       (b) Case from OSAtlas-7B

Figure 18: Layout Grounding error case. Query: "Menu Bar: this region contains the main menu options for the application, including file, edit, view, and more." The red is the ground truth. blue box is the model prediction. The platform of the example is **LibreOffice Impress**.

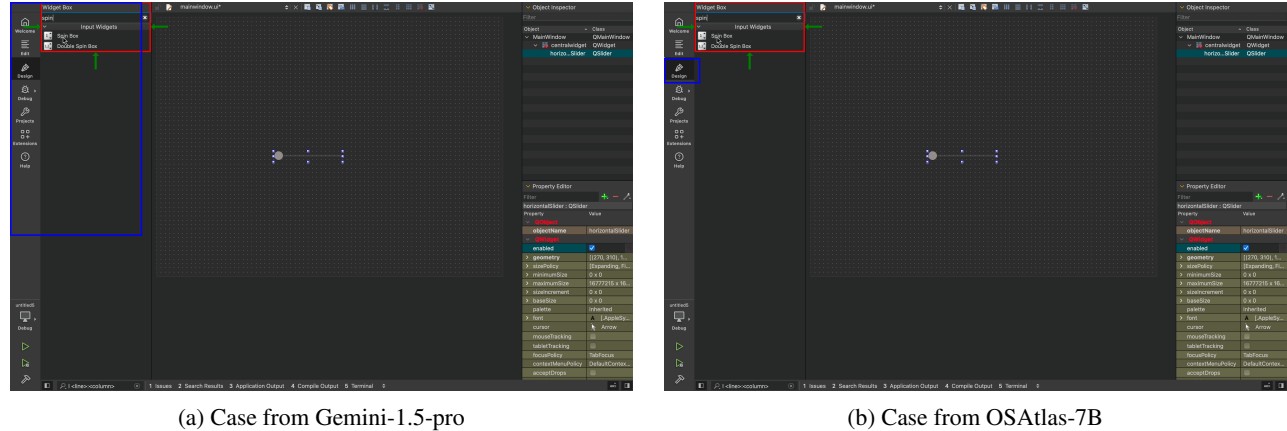

(a) Case from Gemini-1.5-pro       (b) Case from OSAtlas-7B

Figure 19: Layout Grounding error case. Query: "Widget Toolbox: This region contains a collection of widgets that can be used to design and build user interfaces, including buttons, sliders, and input fields." The red is the ground truth. blue box is the model prediction. The platform of the example is **Qt Creator**.

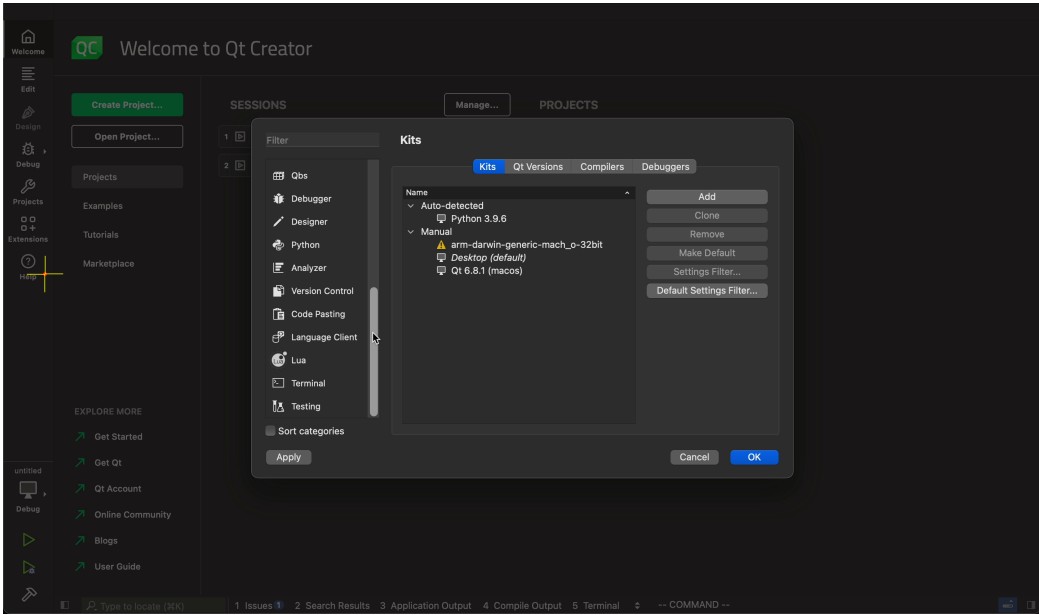

Figure 20: Gemini was instructed to *"Increase the parallel job count to 5 in analyzer settings."* At this step, it correctly predicts the action: Click on Analyzer in the left sidebar menu to open the settings. However, as shown by the yellow arrow, it fails to ground the action in the correct location, clicking on an incorrect element instead. This highlights a common issue where models generate accurate plans but struggle with precise execution.

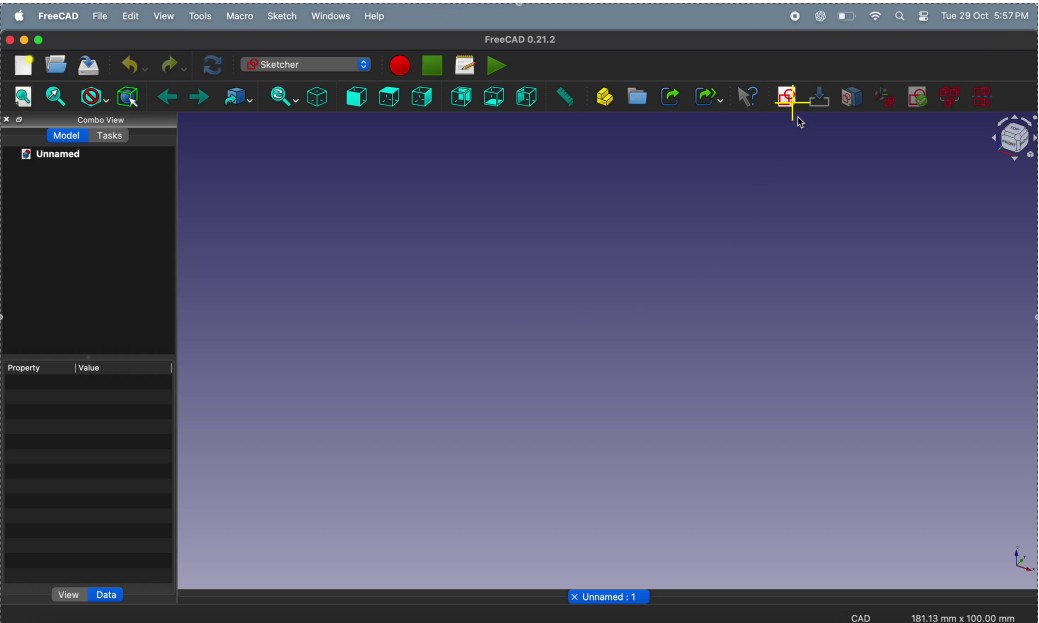

Figure 21: Gemini was instructed to *"In the Sketcher workbench of FreeCAD, set the orientation offset to 10, and draw a 2D ellipse."* However, instead of clicking the Sketcher button (indicated by the arrows in the figure), it predicts the action *"Click on the new sketch button to create a new sketch."* Since a sketch is already present, this action is unnecessary and incorrect. This suggests that Gemini may lack knowledge of the Sketcher icon and its intended functionality.

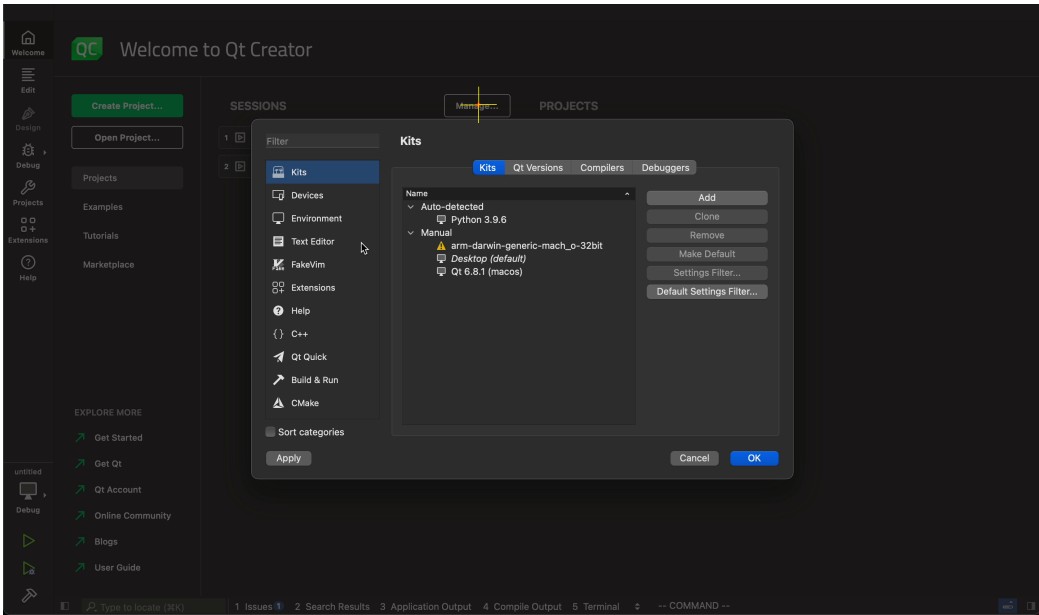

Figure 22: UI-TARS was instructed to *"Enable timeout for 60 seconds for testing."* Instead of dragging through the dialog box's left menu to locate the Timeout setting, it predicts the action *click on the "Manage" button.* Although the action is accurately grounded (yellow arrows on the screen), it is incorrect because the button is in the background and inaccessible without closing the dialog. This suggests that UI-TARS may lack familiarity with the Qt Creator platform and does not recognize the need to scroll within the dialog to find the testing option.

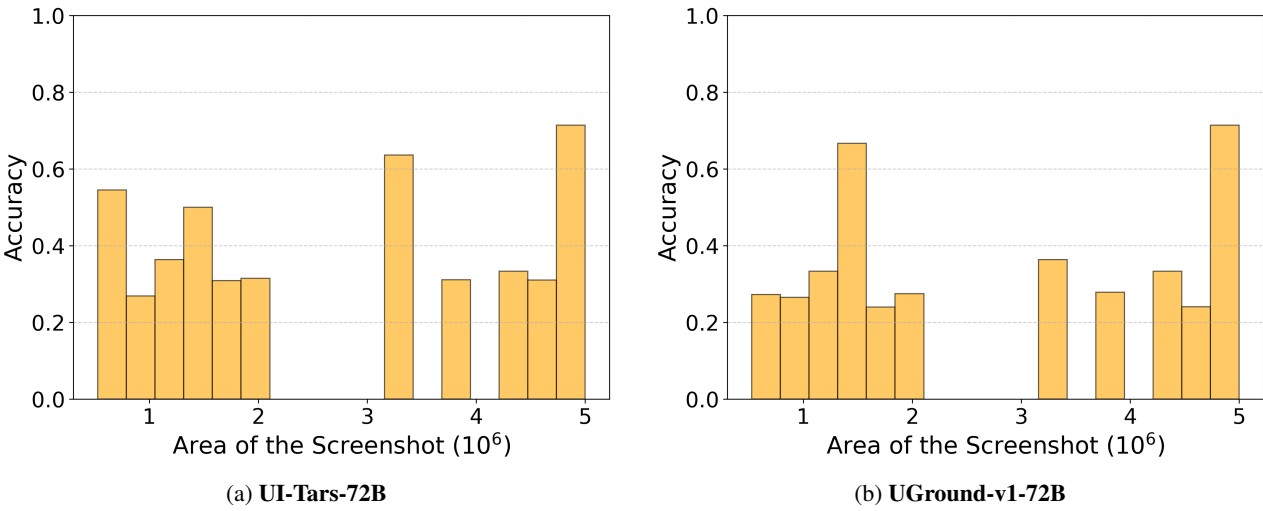

(a) **UI-Tars-72B**

(b) **UGround-v1-72B**

Figure 23: Analysis on Element Grounding accuracy on top two performing models in terms of the area of screenshot (i.e. resolution of screenshot).

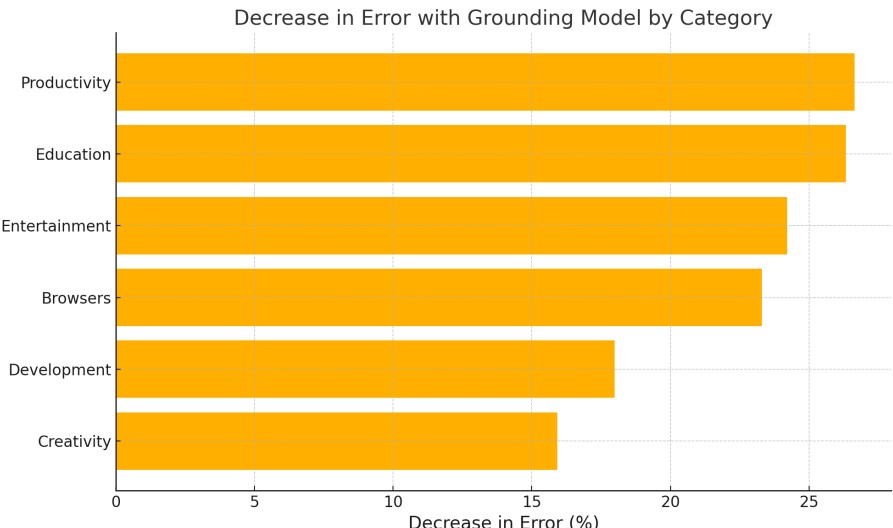

Figure 24: Percentage decrease in error when using GPT-4o as a planner, paired with UGround-v1-7B for grounding the predicted action and target UI element.

