# OpenReview forum: "UI-Vision: A Desktop-centric GUI Benchmark for Visual Perception and Interaction"
_ICML.cc/2025/Conference — ICML 2025 poster_

### Official Review · Reviewer_aEXy · 2025-03-08

**Overall Recommendation:** 3

**Summary:**

The paper introduces UI-Vision, a large-scale, desktop-centric benchmark for evaluating Graphical User Interface (GUI) agents in visual perception and interaction. Unlike existing benchmarks that focus on structured web and mobile interfaces, UI-Vision targets desktop environments, which lack standardized automation APIs and require direct screen interpretation. The benchmark spans 83 open-source applications, providing 6,484 tasks with dense human-annotated data across three key tasks: Element Grounding, Layout Grounding, and Action Prediction. Evaluation of leading Vision-Language Models (VLMs) and GUI agents highlights major limitations, such as poor spatial reasoning (best model achieves only 18.0% accuracy in element grounding) and difficulty in action execution, particularly dragging actions. However, combining LLMs with GUI-grounding models significantly improves action recall (2×–5× gains). The findings emphasize the need for better multimodal training, stronger screen parsing, and enhanced planner-grounding integration for AI-driven GUI interaction. To advance research, UI-Vision is fully open-source, aiming to set a new standard for desktop GUI automation.

**Claims And Evidence:**

1. The paper presents quantitative results showing that state-of-the-art models (GPT-4o, Gemini, etc.) achieve low accuracy on grounding and action tasks. However, the specific reasons for failure (e.g., lack of spatial reasoning, reliance on textual cues) are asserted rather than deeply analyzed. Additional qualitative error analyses or ablation studies could better support this claim. In addition, some new models including Qwen2.5-VL, InternVl-2.5, Cluade-3.7, etc, should be evaluated.

2. While the dataset covers a broad range of applications, it focuses primarily on open-source software, which may not fully capture the complexities of proprietary desktop environments (e.g., Windows/MacOS enterprise applications). The assumption that findings generalize across all GUI platforms could be problematic.

3. The experiments show a 2×–5× improvement in recall when LLMs are paired with grounding models. However, it is unclear whether this improvement is due to better action selection or simply improved recall on basic UI elements. Further analysis of failure cases and different LLM planner strategies would clarify the robustness of this conclusion.

**Essential References Not Discussed:**

None.

**Ethical Review Concerns:**

None.

**Ethical Review Flag:**

Flag this paper for an ethics review.

**Ethics Expertise Needed:**

["Privacy and Security"]

**Experimental Designs Or Analyses:**

1. The data collection process involved human annotators, which could introduce variability. However, the multi-stage quality checks and periodic verification by separate annotators and authors aim to mitigate this.

2. The results indicate that even state-of-the-art models struggle with spatial grounding and precise action execution. This highlights the need for further research and development in this area.

**Methods And Evaluation Criteria:**

1. While UI-Vision is license-permissive, it excludes proprietary applications (e.g., Microsoft Office, Adobe Photoshop, enterprise software). Desktop automation is often used in closed-source environments, so the benchmark might not fully reflect real-world constraints where API restrictions and security settings affect interactions. A more balanced dataset, incorporating at least some closed-source applications, would improve applicability.

2. The Action Prediction task assumes step-by-step task execution, but real GUI agents often require long-term reasoning (e.g., navigating through multiple menus before completing a task). Introducing multi-step action planning evaluation would be beneficial, as current metrics mostly assess immediate action correctness rather than goal completion efficiency.

3. UI-Vision is focused exclusively on desktop applications, whereas many GUI automation challenges also involve cross-platform scenarios (e.g., web-based applications embedded in desktop environments). Incorporating some hybrid environments (e.g., web-based GUIs within desktop software) would better reflect modern GUI automation challenges.

**Other Comments Or Suggestions:**

1. In table 3, based on the performance of open-source VLMS, why InternVL2-8b achieves such low performance in terms of basic setting?

2. Can authors provide more failure case analysises for VLMs? It is important to understand the significance of the proposed benchmark.

**Other Strengths And Weaknesses:**

I hope authors provide anonymous repo containing full benchmarsk and eval scripts. This is extremely important for reviewers to valid the quality of the proposed benchmark.

**Questions For Authors:**

None.

**Relation To Broader Scientific Literature:**

The paper's evaluation reveals specific limitations in current models' spatial understanding and action execution capabilities. It highlights critical areas for future research that align with growing recognition in the literature of the need for improved visual-spatial reasoning in multimodal agents.
In summary, UI-Vision advances the field by addressing documented limitations in existing benchmarks while building on recent progress in multimodal AI and GUI automation research. Its comprehensive approach provides a new standard for evaluating desktop GUI agents, filling important gaps identified in prior literature.

**Theoretical Claims:**

This work focuses on introducing a new benchmark for evaluating Graphical User Interface (GUI) agents, specifically designed for desktop environments. The paper discusses the creation of the UI-Vision benchmark, its tasks, and the evaluation of various models on this benchmark.

---

> ### Author Rebuttal · Authors · 2025-04-01
>
> **R1: Failures cases**
>
> Claude 3.7 Sonnet was released on Feb 24, 24 days after the deadline, and Qwen 2.5-VL on Jan 28, just two days prior—making it infeasible to include them in the review version. However, we have now evaluated both models on our grounding benchmarks, and results are available at <[link](https://bit.ly/4iLiu3j)>.
>
> We summarize key error patterns below and confirm that they hold for the above models and refer the reviewer to our responses to Reviewer 3NZB (R3) and Reviewer 7jPV (R1) for more details and examples:
>
> **Element Grounding:** Models struggle with visually similar elements, platform-specific icons, and small elements in dense UIs. GUI agents perform better on functional tasks due to training alignment, while VLMs underperform, revealing task-specific gaps.
>
> **Layout Grounding:** Frequent issues include overly large or loosely defined bounding boxes and failure to group related UI elements correctly.
>
> **Action Prediction:** Models often fail to ground actions accurately, hallucinate elements/actions, and perform poorly on dense, complex interfaces.
>
> **R2: Concern about generalizing findings from open-source to proprietary desktop environments**
>
> We appreciate the reviewer’s concern. However, many open-source apps closely emulate proprietary counterparts, capturing similar complexity. For instance, LibreOffice mirrors Microsoft Office features and serves over 200 million users[1]. Given this overlap in functionality and interface design, we believe our benchmark remains meaningful and representative of real-world GUI scenarios.
>
> [1]Wikipedia page for LibreOffice. https://en.wikipedia.org/wiki/LibreOffice
>
> **R3: Limited real-world applicability due to the exclusion of closed-source applications**
>
> We agree that automation is widely used in proprietary desktop environments; however, including such applications poses legal and licensing challenges, as many restrict redistribution of software interfaces, recordings, or interaction data, making it difficult to build a publicly shareable dataset around them. To ensure accessibility and reproducibility, we focus on open-source software.
>
> More importantly, our benchmark includes widely-used open-source applications like LibreOffice, VSCode, GIMP, Firefox, Brave, and Shotcut, which serve as strong counterparts to commercial software. These applications support complex, real-world workflows, share high functional similarity with proprietary software and are backed by active communities. We believe this focus ensures UI-Vision remains both practical and widely applicable.
>
> **R4: Planner + VLM grounding analysis**
>
> We clarify that the planner selects the action and target element, while the grounding model only provides coordinates. Thus, the improvement is primarily due to better action selection. To analyze this further, we compared the decrease in error rates across platform categories (Fig. <[link](https://bit.ly/3RimGMd)>) in this setup. Productivity tools show the largest improvement (26%), even though Entertainment tools have the highest baseline grounding accuracy—highlighting the planner’s effectiveness. In contrast, Creativity platforms show the smallest improvement (14%), reflecting the challenges of planning and grounding in functionally rich interfaces with small UI elements.
>
> **R5: Lack of multi-step planning evaluation**
>
> We agree that long-term planning is important, but current models still struggle with single-step actions (nearly 0% recall on drag and only 20% on click actions). Strengthening performance on these core tasks is a necessary first step, with multi-step planning as a valuable direction for future work.
>
> **R6: Missing support for hybrid or cross-platform GUI environments**
>
> We agree that hybrid environments are a valuable direction for future work. However, our current benchmark already presents significant challenges for existing models showing that even desktop settings require substantial progress. We believe UI-Vision offers ample opportunities to advance model capabilities.
>
> **R7: Anonymous repo with full benchmark**
>
> We apologize for the inconvenience but as per ICML policy, we are only permitted to share links to figures and tables. However, to give reviewers a clearer sense of the benchmark, we have included detailed failure cases in our response to Reviewer 3NZB (R3) and provided additional task examples at <[link1](https://bit.ly/4clZMx6)> and <[link2](https://bit.ly/3G06lsO)> (5s loading). All code, data, and evaluation scripts will be soon released.
>
> **R8: Low performance of InternVL-8B**
>
> The main reason is that InternVL2-8B fails to consistently generate meaningful bounding boxes, often producing arbitrary outputs like [0, 0, 50, 50] or [0, 0, 200, 200]. This suggests limited training on grounding tasks, particularly for small UI elements. Its poor performance is also reflected in the ScreenSpot benchmark [1] (Table 2).
>
> [1] Wu, et al. OS-ATLAS: A foundation action model for generalist GUI agents.

---

### Official Review · Reviewer_7jPV · 2025-03-10

**Overall Recommendation:** 2

**Summary:**

This paper introduces UI-Vision, a benchmark for evaluating AI agents’ ability to interact with desktop Graphical User Interfaces (GUIs). Unlike existing benchmarks that focus on web or mobile environments, UI-Vision is designed specifically for desktop platforms and is claimed to be the largest of its kind. It includes 6,484 tasks across 83 software applications, spanning categories such as productivity, development, creativity, education, browsers, and entertainment. The benchmark assesses models on three key tasks: element grounding, layout grounding, and action prediction. The study evaluates multiple state-of-the-art models, including GPT-4o, Gemini, Claude, and various open-source alternatives, highlighting their limitations in handling complex desktop interactions.

**Claims And Evidence:**

The paper claims that UI-Vision is the largest and most diverse benchmark for evaluating desktop GUI agents. This is well-supported by the dataset size (6,484 tasks, 83 applications) and comparisons with existing benchmarks. Claims about performance gaps in SOTA models are backed by experiments.

**Essential References Not Discussed:**

N/A, no obvious omissions were identified from closely related literature.

**Experimental Designs Or Analyses:**

The experimental setup is generally sound, using diverse state-of-the-art models (GPT-4o, Gemini, Claude, and open-source alternatives). The dataset is well-documented, and comparisons with existing benchmarks are thorough. However, there are limitations in the experimental design: 1) The effect of dataset bias, e.g., open-source software selection, is not discussed; 2) No cross-software generalization analysis, e.g., how well models trained on some applications transfer to unseen ones; 3) The study lacks enough error analysis, e.g., why do models fail at certain tasks?

**Methods And Evaluation Criteria:**

The evaluation criteria align well with the problem: GUI automation requires visual perception and interaction, and the paper assesses models across grounding, layout recognition, and action prediction. The dataset is large and diverse, covering real-world applications, which enhances credibility. However, the evaluation could be improved by including human performance baselines to contextualize model failures. The choice of IoU, recall@d, and accuracy metrics makes sense, though more fine-grained analysis (e.g., error types in action prediction) would be useful.

**Other Comments Or Suggestions:**

I look forward to the open-source release of the dataset and resources, which will allow for a more comprehensive review of the benchmark’s reproducibility, cross-benchmark comparisons, and dataset extensibility.

**Other Strengths And Weaknesses:**

Strengths:
•	The first large-scale benchmark specifically designed for desktop GUI automation, addressing a gap in existing benchmarks that focus on web and mobile platforms.
•	Evaluates models on element grounding, layout recognition, and action prediction, providing a multi-faceted assessment of GUI interaction.
•	The dataset spans 83 diverse applications across various categories (e.g., productivity, development, creativity), enhancing its applicability to real-world use cases.
•	Benchmarks state-of-the-art multimodal models, revealing critical performance gaps in GUI perception and interaction.
Weaknesses:
•	Dataset bias: The dataset focuses exclusively on open-source software, limiting its applicability to commercial tools.
•	Limited analysis of model failures: While performance limitations are discussed, there is no in-depth analysis of specific error patterns (e.g., types of misclicks, confusion between similar UI elements, or issues in handling dynamic interfaces).
•	The benchmark focuses on accuracy but does not assess inference speed, latency, or token efficiency for real-world deployment.
•	While the paper evaluates multiple models on UI-Vision, it does not include ablation studies to analyze which task components contribute most to difficulty. Similarly, there is no systematic analysis of how models generalize across different software categories, such as productivity vs. creative tools. Such studies could provide deeper insights into model weaknesses and potential benchmark improvements.

**Questions For Authors:**

1.	If models were previously trained on applications similar to those in UI-Vision, how do you ensure the benchmark fairly evaluates generalization rather than memorization?
2.	Your experiments reveal significant performance drops in spatial element grounding (best model: 18%) and drag actions (near-zero recall). Have you analyzed why these tasks are particularly difficult?
3.	Given that the best-performing model (Gemini-1.5-Pro) achieves only 30.8 IoU in layout grounding, what are the most common failure cases? Do models struggle more with complex, nested UI layouts (e.g., tabbed interfaces) or dense interfaces with many overlapping elements?
4.	Do you anticipate domain shift issues between open-source and closed-source UI designs?
5.	Do you envision automated UI annotation tools or self-supervised learning techniques to reduce reliance on human annotators? What challenges do you foresee in ensuring annotation consistency at scale?

**Relation To Broader Scientific Literature:**

The paper correctly situates UI-Vision within the broader field of GUI automation, multimodal learning, and AI-driven interaction systems. It builds on prior GUI benchmarks like MiniWoB++, WebArena, and OmniAct but extends focus to desktop environments. It also connects with multimodal learning research, citing relevant vision-language models.

**Theoretical Claims:**

The paper does not include formal theoretical claims or proofs. The methodology is empirical, focusing on dataset construction, benchmarking, and performance analysis. No verification of proofs is required.

---

> ### Author Rebuttal · Authors · 2025-04-01
>
> We thank the reviewer for the comments and address them below.
>
> **R1: Fine-grained error analysis, ablations on task difficulty, and generalization across software categories**
>
> To deepen our understanding of model performance, we conducted an error analysis through a human study for element grounding and found that models often struggle with small UI elements and platforms with high functional density. To investigate further, we leveraged the dense bounding box annotations in our dataset and sampled a diverse subset of elements that were consistently challenging for top-performing models such as UI-TARS, UGround-v1, and Aria UI. We selected cases where one or more models failed, resulting in a subset of 5479 samples spanning basic, functional, and spatial categories. We report detailed results across software and categories in Table in <[link](https://bit.ly/4iLiu3j)>, and summarize key findings below:
>
> **Error Types:** Models often confuse visually similar elements, miss platform-specific icons, and fail to detect small elements in dense layouts.
>
> **Task Difficulty:** GUI agents perform better on functional grounding than basic grounding, likely due to alignment with their training data. In contrast, VLMs underperform functional tasks, revealing task-specific weaknesses.
>
> **Category Generalization:** Performance varies significantly across software categories. Creativity tools like Blender and GIMP (112 elements/frame, 418 px/element) show the lowest accuracy, while simpler platforms like VLC (63 elements/frame, 875 px/element) perform best. Notably, screenshot resolution had little impact.
>
> We refer the reviewer to our response to Reviewer 3NZB (R3) for qualitative study and analysis on **Layout Grounding** and **Action Prediction** and more details and examples on **Element Grounding**.
>
> **R2: cross-software generalization analysis**
>
> We compare model performance on common apps (e.g., VSCode) vs. less common ones (e.g., FreeCAD, QGIS) using the element grounding subset in R1. results are included in Table in <[link](https://bit.ly/4iLEnQ5)>. While we cannot confirm exact training data used in several models, this serves as a proxy for generalization analysis. We observe all models show significant accuracy drops on less common apps, confirming consistent generalization challenges.
>
> **R3: Concern about dataset bias due to focus on open-source software**
>
> Many open-source applications closely emulate proprietary counterparts, capturing similar complexity. For instance, LibreOffice mirrors Microsoft Office features and serves over 200 million users[1]. Given this overlap in functionality and interface design, we believe our benchmark remains meaningful and representative of real-world GUI scenarios. We refer the reviewer to our response to Reviewer aEXy (R3) for more details on our choice to focus on open-source software
>
> [1]Wikipedia page for LibreOffice. https://en.wikipedia.org/wiki/LibreOffice
>
> **R4: Evaluation of inference speed, latency and token efficiency**
>
> We perform a detailed analysis of the inference speed, token efficiency and latency for different models and different benchmark tasks and report the numbers in <[link](https://bit.ly/4jeHdNC)>.
>
> **R5: Ensuring generalisation vs memorisation**
>
> Since we do not have access to the training data recipe of the most of models, we are not able to carry out a comprehensive study on this point. However, the failure case of Element Grounding in Fig. 12 in <[link](https://bit.ly/43xRn7f)> indicates merely memorization can not ensure accurate grounding. Models need good generalization ability to excel on the task.
>
> **R6: Analysis of spatial grounding and drag actions**
>
> In the spatial setting, models must first correctly identify the reference element and then reason about its spatial relation—both steps are required for success. Also, VLMs are known to struggle with spatial reasoning, limiting performance.
>
> For drag actions, models are rarely trained on such interactions in web data, making them difficult to execute. Also, success depends on accurately predicting both start and end points, increasing the chance of error.
>
> **R7: Failure cases in layout grounding**
>
> As shown in Fig. 13(a) and 14(a) in <[link](https://bit.ly/4iNTBnL)>, Gemini-1.5-Pro often fails to return a minimal bounding box for the ground truth region, although the correct region is usually contained within the predicted box.
>
> **R8: Domain shift between open and closed software**
>
> Yes, we do. However, open-source systems are built to provide functionalities similar to those of closed-source counterparts, so evaluating model capabilities in these scenarios will directly correlate with those of closed-source ones.
>
> **R9: Potential for automated annotation**
>
> Yes, we do. We have applied LLMs in the annotation of layout grounding in Sec 3.2. However, the major challenge is that UI tasks are quite fine-grained, so it will be hard to control the quality during automatic annotation at scale.

---

### Official Review · Reviewer_PYJp · 2025-03-13

**Overall Recommendation:** 3

**Summary:**

The authors introduce UI-Vision, a comprehensive desktop GUI benchmark with 83 open-source applications, focusing on three tasks: Element Grounding, Layout Grounding, and Action Prediction. Built from human demonstrations and expert annotations, it evaluates GUI agents’ visual perception and interaction capabilities. Tests on top VLMs reveal poor spatial grounding (e.g., 18% accuracy) and action execution (e.g., 4.4% recall on clicks), highlighting gaps in desktop GUI automation.

**Claims And Evidence:**

- UI-Vision’s focus on desktop GUIs fills a critical gap, with its diverse tasks (e.g., layout grounding) offering a fresh approach beyond web/mobile benchmarks.
- Fig. 1 shows a task example but lacks failure cases to illustrate model struggles.
- Sec. 3.2: Layout grounding generation via LLAMA-3.3-70B is mentioned, but validation process details are missing.

**Essential References Not Discussed:**

N/A

**Experimental Designs Or Analyses:**

- Evaluations (Tables 3-5) expose clear model weaknesses (e.g., spatial grounding at 18%, drag action struggles), providing actionable insights for future development.
- No ablations test the impact of annotation density (e.g., 71 boxes/frame) or task design (e.g., spatial vs. functional grounding). How do these affect performance?
- Sec. 3.1: How are "expert annotators" qualified beyond degrees? Training details are vague, affecting reproducibility.

**Methods And Evaluation Criteria:**

- The open-source, densely annotated dataset (83 apps, 2,072 tasks) with real-world complexity is a valuable resource for GUI agent research.
- Table 5 lacks latency metrics, vital for practical GUI automation, despite extensive action evaluation.
- Sec. 4.2: Action metric definitions (e.g., Recall@d) lack specific $d$ values, muddying interpretation.

**Other Comments Or Suggestions:**

N/A

**Other Strengths And Weaknesses:**

- I understand that this is a Desktop-centric GUI Benchmark, but now many benchmarks or datasets [1-4] have already covered these, and I think the unique contribution of this work is a bit limited.

[1] Xu Y, Wang Z, Wang J, et al. Aguvis: Unified Pure Vision Agents for Autonomous GUI Interaction[J]. arXiv preprint arXiv:2412.04454, 2024.

[2] Xie T, Zhang D, Chen J, et al. Osworld: Benchmarking multimodal agents for open-ended tasks in real computer environments[J]. Advances in Neural Information Processing Systems, 2024, 37: 52040-52094.

[3] Zheng B, Gou B, Kil J, et al. GPT-4V (ision) is a Generalist Web Agent, if Grounded[C]//International Conference on Machine Learning. PMLR, 2024: 61349-61385.

[4] Koh J Y, Lo R, Jang L, et al. VisualWebArena: Evaluating Multimodal Agents on Realistic Visual Web Tasks[C]//Proceedings of the 62nd Annual Meeting of the Association for Computational Linguistics (Volume 1: Long Papers). 2024: 881-905.

**Questions For Authors:**

N/A

**Relation To Broader Scientific Literature:**

Good to discuss recent work.

**Theoretical Claims:**

N/A

---

> ### Author Rebuttal · Authors · 2025-04-01
>
> We thank the reviewer for recognizing the value of our desktop-focused benchmark, task diversity beyond web/mobile settings, dense annotations, and actionable insights. We address the concerns below.
>
> **R1: Failures cases**
>
> We summarize key error patterns below and refer the reviewer to our response to Reviewer 3NZB (R3) for more details and examples.
>
> - **Element Grounding:** Models often confuse visually similar elements, fail to recognize platform-specific icons and struggle with small elements in dense layouts.
>
> - **Layout Grounding:** Common issues include bounding boxes that are too large or loosely defined and failure to group related UI elements correctly
>
> - **Action Prediction:** Models frequently fail to ground actions correctly, hallucinate UI elements, and perform poorly on complex interfaces with many interactive elements.
>
> **R2: LLAMA-3.3-70B layout grounding validation process**
>
> The authors conducted a manual validation process to ensure quality using three criteria: **(i)** bounding boxes must tightly enclose relevant UI elements without including unrelated regions; **(ii)** grouped elements must form a semantically meaningful and visually coherent unit; and **(iii)** labels and descriptions must accurately reflect the group's function. Groups failing any check were discarded. The authors had a detailed protocol and access to the platform and its documentation to ensure consistency. We will include these validation details in the camera-ready version.
>
> **R3: Latency metrics**
>
> We report latency per query, average output tokens, and GPU usage across all three tasks in Tables at  <[link](https://bit.ly/4jeHdNC)>, using default Hugging Face implementations for consistency. Token efficiency was measured with GPT-4 tokenization.  Models like UI-TARS, trained on action-heavy tasks, generate longer outputs due to detailed step-by-step reasoning.
>
> **R4: *Recall@d metric and value selection***
>
> We choose the *d* values based on the average bounding box size across the dataset after resizing all screenshots to a standard resolution (800×700) yielding a base value of (25, 35). This base value is then rescaled based on the original resolution of each sample to ensure consistent evaluation across varying interface sizes.
>
> **R5: Ablation on annotation density and task design impact**
>
> We perform a detailed ablation analysis (Fig. 18 at <[link](https://bit.ly/4c8La3M)>) to understand factors affecting performance on 3 tasks. Across all three tasks, we find that densely packed applications with smaller UI elements like GIMP (112 elements/frame, area of 418 px/element) show lower performance compared to entertainment platforms with simpler layouts like VLC (63 elements/frame, area of 875 px/element).
>
> Regarding task design, GUI agents perform comparably or better on functional grounding than basic grounding, likely due to alignment with their training data. In contrast, both open- and closed-source VLMs perform worse on functional tasks. Spatial grounding is the most challenging, as it requires identifying the correct element and reasoning about its relative position—an area where VLMs generally struggle due to limited spatial reasoning.
>
> For a more comprehensive analysis with detailed settings, we refer the reviewer to our detailed response to Reviewer 7jPV (R1).
>
> **R5: annotator qualifications and training process**
>
> Beyond academic background, annotators were selected through technical assessments, language tests, and task-specific bootcamps. Those who didn’t meet the criteria were excluded. We used a detailed annotation protocol refined during a month-long pilot with feedback to ensure consistency (L-676–677). Quality was further ensured through manual reviews and ongoing performance monitoring. We will clarify these details in the camera-ready version.
>
> **R6: Clarification on UI-Vision’s unique contribution vs. existing benchmarks**
>
> We appreciate the reviewer’s concern and would like to clarify the distinct contribution of our work. While recent efforts [1–4] have advanced GUI agent evaluation, they primarily focus on web environments or online interaction. Specifically, [1] includes limited desktop data and annotations (OmniAct: 5412 training samples across 38 platforms only for action prediction). [3] and [4] focus mostly on web-based tasks. While [2] (OSWorld) targets desktop platforms, it evaluates across a limited number of platforms in an online setting using broad task completion metrics.
>
> In contrast, our benchmark supports offline evaluation across 450 real-world desktop tasks spanning 83 applications. It provides a complete pipeline of three benchmark tasks, along with detailed evaluation metrics. This setup allows models to be assessed from basic perception to planning and execution, all within a single structured benchmark. Unlike existing works, UI-Vision offers fine-grained insights into where and how models fail, making it a valuable tool for diagnosing and improving GUI agents.

---

### Official Review · Reviewer_3NZB · 2025-03-13

**Overall Recommendation:** 3

**Summary:**

This paper introduces a desktop GUI benchmark (i.e., UI-Vision) that spans 83 real-world environments with open-source and permissive data. It enables three key tasks evaluation, including element grounding, layout grounding, and action prediction. The evaluation reveals the limitations of existing works to handle desktop environments.

## update after rebuttal

I appreciate the authors' clarifications. My main concerns have been addressed by the rebuttal. I would lean to accept the paper by involving the additional discussions in the revised version.

**Claims And Evidence:**

1. Claiming that the proposed benchmark is the largest desktop-centric benchmark is a little bit unconvincing. As shown in Table 1, the proposed benchmark contains 6484 samples, while OmniAct (Kapoor et al., 2024) contains 9802 samples.
2. L209 states that the final dataset consists of 442 high-quality demonstrations across 83 applications. But L055 (right) states that the proposed method contains 450 recorded videos spanning 83 platforms.

**Essential References Not Discussed:**

No.

**Experimental Designs Or Analyses:**

It would be better to add qualitative evaluations to the paper.

**Methods And Evaluation Criteria:**

The proposed benchmark dataset is useful for the comprehensive evaluation of autonomous GUI agents, covering essential agent capabilities.

**Other Comments Or Suggestions:**

The number of videos is inconsistent in the paper.

**Other Strengths And Weaknesses:**

The paper is well-organized with a clear structure. The proposed benchmark dataset could be useful for future research in the community. However, this benchmark still belongs to offline and static scenarios.

**Questions For Authors:**

How to ensure the consistency of human annotators during the dataset creation process?

**Relation To Broader Scientific Literature:**

The key contribution of this paper is the proposed desktop-centric GUI benchmark. It is useful for comprehensive element grounding, layout grounding, and action navigation tasks.

**Theoretical Claims:**

Yes.

---

> ### Author Rebuttal · Authors · 2025-04-01
>
> We thank the reviewer for recognizing the value of our desktop-centric benchmark, its utility for comprehensive analysis of GUI agents, and the clarity of the paper. Below, we address the concerns raised.
>
> **R1: Clarification on “largest desktop-centric benchmark” claim**
>
> We appreciate the reviewer’s point and agree that “largest” can be interpreted in different ways. While OmniAct includes 9802 samples (7639 desktop-related), it focuses solely on action prediction. In contrast, our benchmark includes 6484 samples across 83 applications and supports three tasks—action prediction, layout and element grounding—with dense frame-level UI annotations. Considering the range of platforms, tasks, and annotation details, we believe UI-Vision is the most comprehensive desktop GUI benchmark, and we are happy to revise the claim for clarity.
>
>
> **R2: Clarification on 442 vs. 450 video count inconsistency**
>
> To clarify, our dataset contains 450 densely annotated videos across 83 applications. However, for the action prediction task, we use 442 videos, excluding 8 that involved complex actions (e.g., press and hold Ctrl + drag + release), which most models cannot yet handle reliably making evaluations difficult. This is noted in the Limitations section (L795–797), and we will ensure both this and the earlier typo are clarified in the camera-ready version.
>
> **R3: Qualitative Evaluation**
>
> Below we highlight qualitative error analysis for all three benchmark tasks:
>
> **Element Grounding:** Figures are available at [](https://bit.ly/43xRn7f)<[link](https://bit.ly/43xRn7f)>. Observations are on SOTA UI-TARS model
> >**Fine-grained ambiguity:** The model fails to recognize the correct target among several visually similar candidates (Fig. 9), highlighting the need for improved disambiguation strategies.
>
> >**Lack of domain knowledge:** The model misinterprets platform-specific elements, such as  “Fontwork” represented by “F” symbol  (Fig. 10). Incorporating external knowledge could help improve performance.
>
> >**Small element detection:** The model struggles with small UI elements, particularly in high-resolution or dense interfaces (Fig. 11). Iterative zoom-in strategies may address this limitation.
>
> >**Cross-platform generalization:** The model incorrectly transfers layout assumptions across platforms—eg., predicting the minimize button's position on macOS as it would appear on Windows (Fig. 12). This suggests memorization and overfitting.
>
> **Layout Grounding:** Example cases below are available at <[link](https://bit.ly/4iNTBnL)>
> >**Inaccurate bounding box placement:** Closed-source models often predict bounding boxes that loosely cover the correct region without precisely matching its boundaries (Fig. 13a, 14a). This suggests difficulty in precise layout partitioning
>
> >**Poor functional grouping:** Open-source models sometimes fail to group elements correctly, even when the query explicitly mentions them (Fig. 13b).
>
> >**Superficial semantic matching:** Models sometimes default to grounding smaller elements that share surface-level keywords with the query but are semantically unrelated (e.g., predicting a “Design” button for a different design-related query, Fig. 14b)
>
> **Action Prediction:** Example cases below are available at <[link](https://bit.ly/42gIBID)>
> >**Poor grounding:** Models often predict the correct action type but fail to ground it to the appropriate UI element (Fig. 15). This reflects challenges in bridging perception and execution.
>
> >**Lack of platform knowledge:** We observe that models sometimes hallucinate actions (eg., refer to non-existent elements) or misinterpret platform-specific elements (Figs. 16, 17), likely due to limited training exposure to diverse desktop environments.
>
> >**High interface complexity:** Dense and feature-rich platforms pose greater challenges for accurate action prediction. UI-TARS exhibits the highest error rate (85%) on creativity platforms (112 elements/frame) while performing better (72% error) on simpler education platforms (62 elements/frame).
>
> **R4: Clarification on offline/static benchmark setting**
>
> Our benchmark is offline and static by design. The controlled setup allows for a fine-grained evaluation of how perception and grounding errors affect downstream actions. By structuring tasks from perception to action prediction, UI-Vision helps isolate failure modes and provides insights crucial for building more robust agents before deployment in dynamic environments.
>
> **R5: Annotator consistency during dataset creation**
>
> We partnered with a for-profit company that provided experienced annotators (L-604-605). All annotators underwent training on the software and were required to pass several assessments related to the tasks to proceed. Those who failed were excluded. Additionally, we followed a detailed annotation protocol, which was refined during a month-long pilot phase where annotators received detailed feedback to ensure high-quality and consistent data (L-676-677).

---

### Decision · Program_Chairs · 2025-05-01

**Decision:**

Accept (poster)

**Comment:**

This paper presents a comprehensive benchmark for understanding complex desktop GUIs. The benchmark features high-quality, human-annotated action trajectories, diverse layout and interaction annotations, and spans 83 software applications to capture broad real-world variability. Experimental results reveal that current models struggle with desktop environments, highlighting the need for further research. After discussion, three reviewers recommended a weak accept, while Reviewer 7jPV maintained a weak reject. The Area Chair acknowledges that some of Reviewer 7jPV’s concerns were addressed in the rebuttal, though questions remain about the suitability of a benchmark-focused paper for ICML. As a result, the Area Chair recommends a weak accept, and encourages the authors to incorporate the necessary revisions in the final version.